# Fingerprints of quantum criticality in locally resolved transport

## Xiaoyang Huang⋆ and Andrew Lucas†

Department of Physics and Center for Theory of Quantum Matter,
University of Colorado, Boulder CO 80309, USA

⋆ xiaoyang.huang@colorado.edu, † andrew.j.lucas@colorado.edu

## Abstract

Understanding electrical transport in strange metals, including the seeming universality of Planckian $T$-linear resistivity, remains a longstanding challenge in condensed matter physics. We propose that local imaging techniques, such as nitrogen vacancy center magnetometry, can locally identify signatures of quantum critical response which are invisible in measurements of a bulk electrical resistivity. As an illustrative example, we use a minimal holographic model for a strange metal in two spatial dimensions to predict how electrical current will flow in regimes dominated by quantum critical dynamics on the Planckian length scale. We describe the crossover between quantum critical transport and hydrodynamic transport (including Ohmic regimes), both in charge neutral and finite density systems. We compare our holographic predictions to experiments on charge neutral graphene, finding quantitative agreement with available data; we suggest further experiments which may determine the relevance of our framework to transport on Planckian scales in this material. More broadly, we propose that locally imaged transport be used to test the universality (or lack thereof) of microscopic dynamics in the diverse set of quantum materials exhibiting $T$-linear resistivity.



# 1  Introduction

The strange metal, which exists at temperatures above the superconducting $T_c$ of the high-$T_c$ superconductors, remains one of the most mysterious phases of quantum matter found in Nature. Most famous among these mysteries is $T$-linear resistivity [1–3], which seems to persist well below the Debye temperature (above which classical phonon scattering gives this result [4]). The absence of particle-like excitations, revealed by photoemission [5,6], suggests that the strange metal is best described by non-quasiparticle theories of strongly correlated electrons. Given the apparent proximity of many strange metals to a quantum phase transition at zero temperature [7,8], it has been conjectured for some time that this $T$-linear resistivity may partially or wholly be a consequence of quantum critical dynamics above a quantum critical point (hidden by superconductivity) [9]. The $T$-linear resistivity then arises as a consequence of a quantum mechanical "bound": the Drude scattering time should obey $\tau \gtrsim \hbar/k_{\mathrm{B}}T$ [10,11]. The saturation of this bound in a strange metal is quantitatively consistent with numerous experiments [2,3,12,13]. Many exotic theories of strange metals, including those based on field theories of quantum criticality [14–16], Sachdev-Ye-Kitaev chains [17,18], and gauge-gravity duality [19,20], have been proposed to elucidate why this Planckian scattering time can ultimately enter the resistivity. Unfortunately, because (in large part) theories based on either standard frameworks (kinetic theory) or non-standard ones (criticality or holography) strive to reproduce the same Planckian $T$-linear resistivity, it has been notoriously challenging to select which (if any) of these theories gives a qualitative and predictive theoretical foundation for strange metallic transport.

Here, we argue that novel experimental techniques, such as scanning single-electron transistors (SET) [21] or nitrogen vacancy center magnetometry (NVCM) [22–24], could be used to reveal quantum critical dynamics (or its absence) in transport experiments on strange metals, by *locally* studying how current flows in response to an applied electric field. After all, ordinary transport measurements report a single number: the resistivity $\rho$, or conductivity $\sigma = 1/\rho$, at a fixed temperature. But a local imaging experiment can (indirectly) return a

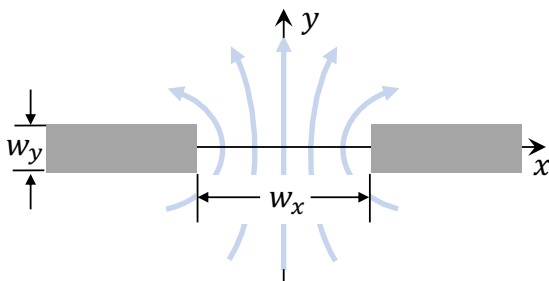

Figure 1: Cartoon of current flow through a constriction; the widths $w_x$ and $w_y$ are depicted. Region O is unshaded; region I (the constriction) is shaded gray. We take $w_x \gg w_y$, so the physics is largely insensitive to the small value of $w_y$.

*function* $\sigma(\boldsymbol{x})$, which relates local current $J_i(\boldsymbol{x})$ to local electric field $E_i(\boldsymbol{x})$ via

$$J_i(\boldsymbol{x}) = \int \mathrm{d}^2 x' \, \sigma_{ij}(\boldsymbol{x} - \boldsymbol{x}') E_j(\boldsymbol{x}'). \tag{1}$$

Here the $i, j$ indices correspond to spatial directions; repeated indices are summed over. Today, the literature contains extensive measurements on the homogeneous part of this equation (uniform current response to uniform electric field), yet very little data in only select materials on the non-local response arising due to the $x$-dependence in $\sigma_{ij}(\boldsymbol{x})$. Knowledge of the whole function $\sigma_{ij}(\boldsymbol{x})$ may reveal a stunning amount of universality between all strange metals, strongly hinting at a universal origin (perhaps arising from quantum criticality); or, it may reveal that Planckian universality is an illusion, with non-universal, material-specific phenomena responsible for $\rho \sim T$ in a strange metal. Directly coupling a metal to electromagnetic waves gives correlators at $\omega = ck$, with $c$ the large speed of light. In order to measure $\sigma(k, \omega \to 0)$, a more indirect approach implementable in present-day experiments is necessary.

## 2 Locally resolved transport

We now explain a method to calculate $J_i(\boldsymbol{x})$, and thus $\sigma(k)$, in an engineered device geometry, which was first proposed and studied in the context of viscous hydrodynamic electron flow in [22, 25]. Consider (1) in the presence of a non-trivial geometry, as depicted in Fig. 1. If we attempt to apply a uniform electric field, the presence of "hard walls" will force current to move around them; the local forcing of these currents will necessarily arise from local electric fields that build up due to space charges rearranging themselves in the metallic leads. So we may write

$$E_j(\boldsymbol{x}) = E_j^{(0)} + \tilde{E}_j(\boldsymbol{x}), \tag{2}$$

where $E_j^{(0)}$ is a constant background field, and $\tilde{E}_j$ denotes the perturbation to the electric field arising due to the device walls. We separate the computational domain to regions I (inside) and O (outside) the "walls" of the device, where we assume current cannot flow: $J_i(\boldsymbol{x} \in I) = 0$. We make the *ansatz* that $\tilde{E}_i(\boldsymbol{x} \in O) = 0$ and

$$\tilde{E}_i(\boldsymbol{x} \in \mathrm{I}) = -\int_{\mathrm{I}} \mathrm{d}^2 x' \left[ b \delta_{ij} \delta_{\boldsymbol{x}, \boldsymbol{x}'} + \sigma_{ij}(\boldsymbol{x} - \boldsymbol{x}') \right]^{-1} \sigma_{jk}^{(0)} E_k^{(0)}, \tag{3}$$

where $\sigma_{jk}^{(0)}$ is the zero wave number Fourier mode of $\sigma_{ij}(\boldsymbol{x})$, and the limit $b \to 0$ is taken. We may now calculate the current flow pattern $J_i$ by combining (1), (2) and (3) to evaluate $J_i(\boldsymbol{x})$ in region O.

We consider (1) to be exact with $\sigma_{ij}(\boldsymbol{x}, \boldsymbol{x}') = \sigma_{ij}(\boldsymbol{x} - \boldsymbol{x}')$, even in the presence of an inhomogeneous device: the "effective" electric field $\tilde{E}_i$ inside of region I (outside the physical domain of the metal) encodes boundary conditions analogously to the method of image charges in electrostatics. The limit $b \to 0$ effectively encodes the condition that "image currents" cancel out the otherwise uniform current imposed by $E_i^{(0)}$. For more detailed discussions of our algorithm, we refer readers to Appendix A. We emphasize the numerical efficiency that the only matrix inversion required to solve the linear systems in (1) and (3) takes place in region I. As long as region I contains $\sim 10^3$ grid points, we can perform the calculation without specialized numerical methods. Note also that this algorithm can be done regardless of the shape of region I, and its complement O; see the appendices.

So far, we have established a framework for calculating current distributions. This can be further generalized to the distributions of an arbitrary operator $\mathcal{O}$ through

$$\mathcal{O}(\boldsymbol{x}) = \mathcal{O}_0(\boldsymbol{x}) + \int \mathrm{d}^2 x' \sigma_{\mathcal{O}J_i}(\boldsymbol{x} - \boldsymbol{x}') \tilde{E}_i(\boldsymbol{x}'), \tag{4}$$

where $\tilde{E}_i$ is the induced electric field and $\sigma_{\mathcal{O}J}$ is the generalized conductivity tensor: see Appendix I for details. The $\mathcal{O}_0$ corresponds to the response to the external constant field $E_i^{(0)}$ and will typically vanish (especially if $\mathcal{O}$ is not a spatial vector operator). In Appendix I, we apply (4) to calculate the bulk charge distribution, $n(\boldsymbol{x})$ or $\mu(\boldsymbol{x})$, in order to determine the total conductance.

## 3 Conductivity

By our assumed translation invariance in (1), it suffices to calculate the Fourier transform

$$\sigma_{ij}(\boldsymbol{k}) = \left( \delta_{ij} - \frac{k_i k_j}{k^2} \right) \sigma(k), \tag{5}$$

where $k^2 = k_x^2 + k_y^2$. Note that current conservation in two dimensions demands that $\sigma_{ij}(\boldsymbol{k})$ can be characterized by a single function $\sigma(k)$, as above. We then calculate the (real) conductivity via the holographic correspondence (see the appendices):

$$\sigma(k) = \lim_{\omega \to 0} \frac{\mathrm{Im}\, G^{\mathrm{R}}_{JJ}(\omega, k)}{\omega} = \lim_{\omega \to 0} \frac{1}{\omega} \mathrm{Im}\, \frac{\partial_r a_y(r = 0)}{a_y(r = 0)}. \tag{6}$$

Here, $G^{\mathrm{R}}_{JJ}(\omega, k)$ is the Fourier transform of the retarded Green's function; $a_y(r = 0)$ corresponds to a fluctuating bulk gauge field, evaluated at the boundary ($r = 0$) of the bulk gravity theory. We note that in principle, one could try to implement a holographic "constriction geometry" and compute exactly $\sigma_{ij}(\boldsymbol{x}, \boldsymbol{x}')$ through it; however, to obtain the spatial resolution of our images by solving the inhomogeneous holographic model, we would require the numerical inversion of matrices with at least $10^5 - 10^6$ rows and columns, since the holographic PDEs would need to be solved in an additional bulk radial direction.

## 4 Zero density

We begin by studying the resulting physics when the charge density $n = 0$. More precisely, our holographic model describes a $2 + 1$-dimensional conformal field theory (CFT), with global U(1) symmetry, studied at finite temperature $T$. Based on very generic arguments, we may

anticipate some of our numerical results. If the constriction is extremely large, then at finite temperature we expect charge transport to be ohmic at long wavelengths:

$$J_i = \sigma_0(E_i - \partial_i \mu), \tag{7}$$

where $\mu$ is the local chemical potential and $\sigma_0$ is the (incoherent) conductivity [11,20]. Note that (7) is a *hydrodynamic* prediction, despite this phrase often meaning viscous finite density transport (which we will observe once $n \neq 0$). Indeed at zero density, the charge degree of freedom does not interact with other hydrodynamic modes (energy and momentum), and therefore the long wavelength physics of charge transport will appear *identical* to textbook ohmic theory. In ohmic transport, the current profile is sharpest near the corners of the constriction. This universal result follows the same mathematics responsible for the blow-up of electric field magnitudes near the sharp corners of a lightning rod [22].

At sufficiently short length scales, hydrodynamics breaks down. Letting $c$ denote the speed of light in the CFT, we estimate that the length scale below which hydrodynamics does not exist is $\ell_{\mathrm{Pl}} = \hbar c / k_{\mathrm{B}} T$ [1]; for simplicity in what follows, we will generally work in units where $\hbar = c = k_{\mathrm{B}} = 1$. For a CFT, this result follows from dimensional analysis: $T$ is the only dimensionful parameter in the theory. Hence, we expect that when $w_x \gg \ell_{\mathrm{Pl}}$, the current distribution looks ohmic, and when $w_x \ll \ell_{\mathrm{Pl}}$, the current distribution does not look ohmic. When $w_x \ll \ell_{\mathrm{Pl}}$, it is natural to expect that the theory looks essentially like a zero temperature CFT. Unfortunately, in this particular problem, the response of a pure CFT is pathological [26, 27]. Current conservation and conformal invariance together imply [26]

$$G^{\mathrm{R}}_{J_\mu J_\nu} = K|p|\left(\eta_{\mu\nu} - \frac{p_\mu p_\nu}{p^2}\right), \tag{8}$$

where $\eta_{\mu\nu} = \mathrm{diag}(-1,1,1)$, $p_\mu = (\omega, \mathbf{k})$, $p = \sqrt{p_\mu p_\nu \eta^{\mu\nu}}$ and $K$ is a constant characterizing the CFT. In the limit $\omega \to 0$ at $k$ fixed, $G^{\mathrm{R}}_{JJ}$ is purely real, and $\sigma(k) = 0$ is predicted by the CFT. Because we are in fact at finite temperature, there will be small corrections to $\sigma(k)$, which depend on the ratio $k/T$ [28]. In holographic models, general arguments [20] imply that

$$\sigma(k) \sim \exp(-\alpha k/T), \tag{9}$$

where $\alpha$ is a theory-dependent constant. Since $\sigma(k)$ decays extremely rapidly with $T$, we qualitatively expect that the longest wavelength current distribution that can fit inside the constriction will dominate the current profile, and therefore predict an approximately sinusoidal profile of current flow through the constriction, with "wavelength" of order $w_x$. This distribution is peaked *away* from the constriction edges, so is easily distinguished from Ohmic transport. We call this sinusoidal current distribution on short length scales *quantum critical*. Details of this argument are in Appendix D.

Thus we have clear predictions: for a fixed constriction width $w_x$, at temperatures $T \gtrsim 1/w_x$, we will see that the current profile through the constriction is peaked at the sides; for $T \lesssim 1/w_x$, it is peaked in the middle. This feature, along with other key predictions above, are precisely observed in our numerical computations of the current distribution, presented in Fig. 2. By comparing to a simple "kinetic" prediction for a Fermi liquid, this plot further justifies that our "fingerprint" actually does discern between two different models of short-distance transport: a quantum critical regime vs. an ohmic regime.

In the appendices, we show that non-interacting Dirac fermions exhibit the same sharply peaked current profile as the holographic model on scales much smaller than the Planckian length scale, due to (9) also being obeyed. This is further evidence for our claim that this is a signature of quantum critical transport. The results of [29] for theories with $z > 1$ suggest that even non-relativistic critical systems may have similar non-local response, in which $\sigma(k)$ exponentially decays with large $k$ (although the Planckian length scale will scale differently with

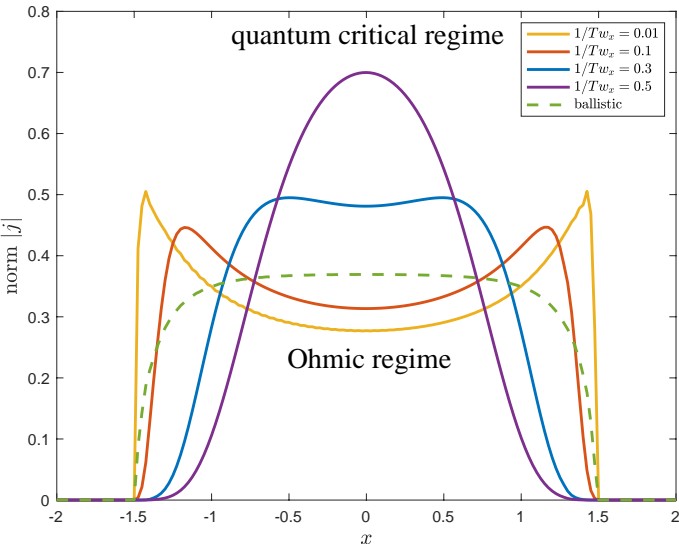

Figure 2: Simulated examples of current flow at zero density through a slit of $w_x = 3$ $\mu$m and $w_y = 0.04$ $\mu$m. The distribution evolves from doubly peaked in the Ohmic regime to singly peaked at the center in the quantum critical regime. The "kinetic" prediction for physics on length scales smaller than Planckian, $\ell_{ee} = \ell_{mr} \gg w_x$ (known as the ballistic limit), is further shown in the dashed green curve.

temperature). Exponential decay in $\sigma(k)$ analogous to (9) will always lead to the sinusoidal current profile, which is our "fingerprint" of quantum criticality.

## 5  Finite Density

Now, let us generalize to finite density $n > 0$ (the sign of $n$ does not matter for what follows). Again, we first state a few generic expectations. As before, at sufficiently long wavelengths, $\sigma(k)$ will be approximately hydrodynamic [20, 30]:

$$\sigma(k) = \sigma_0 + \frac{n^2}{\eta k^2}. \tag{10}$$

The above expression has both an incoherent conductivity $\sigma_0$, and a "coherent" piece which arises from the overlap of charge current and momentum (which is universal at finite density) [20]. Generically, $n$, $\eta$ and $\sigma_0$ are all functions of $T$ and $\mu$, which may explicitly be calculated holographically. However, the key prediction of (10) is that for sufficiently large wavelength, the current profile will be dominated by the viscous term. The viscous current profile through a constriction with our boundary conditions is known [22, 25, 31] to be semi-circular: like the quantum critical regime, the current has a maximum in the middle of the constriction; however, it is also much less sharply peaked.

Qualitatively, our predictions for the quantum critical regime are identical to before. Quantitatively, it is not necessary for (9) to hold, as the precise form of spectral weight will depend on details of the low energy theory. Holographically, this low energy regime (when $\mu \gg T$) is known to exhibit *local quantum criticality* [32]; see the appendices. As before, we have numerically confirmed this prediction in Fig. 3.

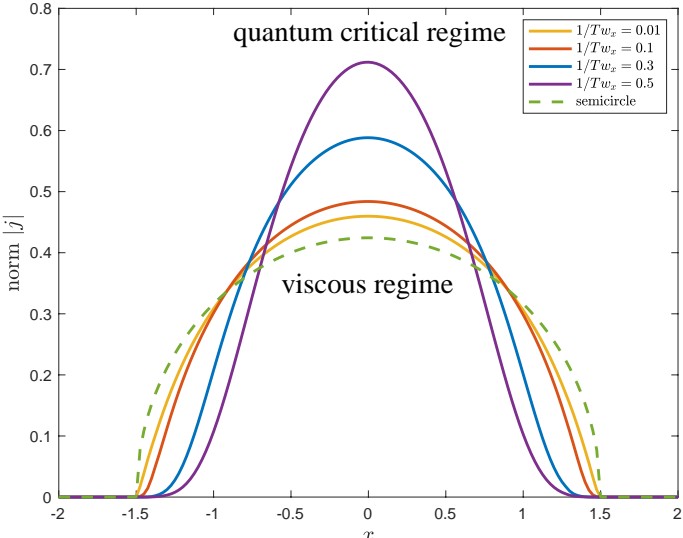

Figure 3: Simulated examples of current flow distribution at finite density ($Q = 0.5$, as defined in the appendices) through a slit of $w_x = 3$ $\mu$m and $w_y = 0.04$ $\mu$m. The distribution evolves from semi-circular in the viscous regime to sinusoidal in the quantum critical regime.

## 6    Comparison to Experiment

The recent experiment [22] imaged current flow patterns in high quality monolayer graphene. At zero density, these authors observed ohmic profiles at all temperature scales; however, modeling the current distributions at charge neutrality by Fermi liquid Boltzmann transport theory may be questioned: charge neutral graphene has no Fermi surface, and an interaction length comparable to the Planckian length scale (suggesting the breakdown of quasiparticles). After all, since the only scale at zero density is $T$, the effective "Planckian" length scale must be [12, 33]

$$\ell_{\mathrm{qc}} \sim \frac{1}{C} \frac{\hbar v_{\mathrm{F}}}{k_B T}, \tag{11}$$

where $v_{\mathrm{F}} \approx 10^6$ m/s is the Fermi velocity, $C$ is a dimensionless number relating $\ell_{\mathrm{qc}}$ to the effective fine structure constant in graphene. Although the experiment [22] was unable to probe physics at both $T \gtrsim 1/w_x$ and $T \lesssim 1/w_x$, as $w_x \approx 3$ $\mu$m while $\ell_{\mathrm{qc}} \sim 300$ nm, we can still compare their data to our theory of charge neutral quantum critical transport to estimate whether the observed *change* in current profiles with temperature is compatible with our model when the parameters are physically realistic.

We chose the effective speed of light $c$ in our holographic model to correspond to $v_{\mathrm{F}}$. Hence, to fit the experimental data at $T = 297$ K and 128 K, we are left with one fit parameter: $C$. Put another way, in our zero density model, the only free parameter to be tuned is the combination $\ell_{\mathrm{qc}}/w_x \sim 1/T_{\mathrm{fit}} w_x$. Here $T_{\mathrm{fit}}$ is defined to be the effective temperature where, assuming $C = 1$ in (11) (consistent with our holographic model which also set $\hbar = v_{\mathrm{F}} = k_{\mathrm{B}} = 1$), the resulting fit best matches the experimentally measured current profile. We find that (in natural units) $1/T_{\mathrm{fit}} w_x \approx 0.05$ and 0.11 for $T = 297$ K and $T = 128$ K, respectively (Fig. 4(a)). Importantly, observe that the ratio of these fitting temperatures is close to the ratio of experimental temperatures, which implies we can meaningfully extract our model's estimate of the dimensionless constant $C \approx 0.18$. The constant $C$ we obtain is very close to the experimental result $C \approx 0.2$ reported in [12], providing a quantitative check on the validity of our approach.

We advocate that future experimental work studies flow of the Dirac fluid through con-

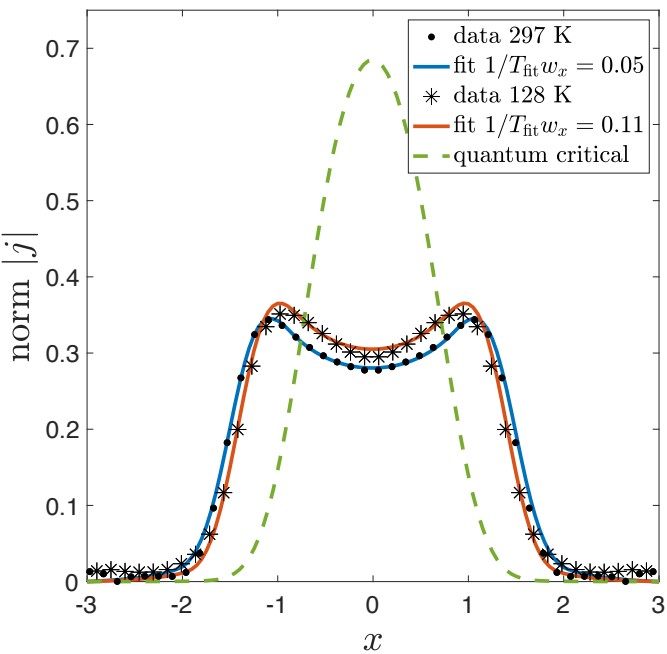

Figure 4: Normalized current profile $|j|$ across the slit of $w_x \approx 3$ $\mu$m at the charge neutrality point at 297 K and 128 K. The solid lines indicate the best fit based on the holographic model, $1/T_{\text{fit}}w_x = 0.05$ and 0.11 for 297 K and 128 K, respectively, Shown in the dashed green curve is the predicted behavior in a constriction of width 600 nm for the experimental temperature at $T = 128$ K. Details of the fitting method are provided in the appendices.

strictions of order 600 nm in width at $T \sim 100$ K; in this regime, and using higher resolution magnetometry, it should be possible to easily distinguish between "Fermi liquid" and quantum critical transport phenomena, as shown in Fig. 4. Of course, even if transport were to look quantum critical, as in Fig. 4 – this may not be sufficient to demonstrate the absence of quasiparticles; more realistic kinetic theories [34, 35] which incorporate both electron and hole dynamics must also be analyzed, and the breakdown of semiclassical dynamics on length scales smaller than $\hbar v_{\text{F}}/k_{\text{B}}T$ must be accounted for. Without doing an exhaustive analysis here, we anticipate that the model of [34, 35] still suggests viscous-like flows due to the emergence of approximate momentum conservation of electron and hole fluids separately; their momentum-dependent conductivity $\sigma(k) \sim 1/(1 + v^2 \tau_{c,1} \tau_{c,2} k^2)$ is quite different from the quantum critical regime, where we have predicted (9).

In experiments on graphene samples using hBN substrates one typically finds that the main source of disorder is inhomogeneity in charge density, which are called charge puddles. The amplitude of charge puddles is around 30 K (namely, the fluctuations in local Fermi temperature of this order), and their size is around 100 nm [30, 36]. Since these numbers are both small relative to what we advocating to detect quantum critical flows, it is likely reasonable to neglect the charge puddles when studying flows in our proposed device. Further experimental work along these lines is warranted.

## 7 Outlook

We have proposed a simple and generic signature for quantum criticality in the spatially resolved transport of strange metals (see Tab. 1). Using state-of-the-art local probes, local trans-

Table 1: Summary of how to distinguish between different qualitative flow regimes.

| | ohmic | ballistic | viscous | quantum critical |
|---|---|---|---|---|
| current profile | | | | |
| conductance | $\log w_x$ | $\sqrt{n}\, w_x$ | $n^2 w_x^2/\eta$ | $\exp(-\tilde{\alpha}/w_x)$ |

port may soon be imaged in the Dirac fluid of graphene [12, 37], magic angle twisted bilayer graphene [38–40], or high-$T_c$ superconductors, where strange metallic behavior is often believed to result from quantum criticality.

The ability to image current flows will unambiguously distinguish between ohmic and non-ohmic flow patterns. Whether or not transport indeed looks ohmic at $\ell_{Pl}$, along with the current distribution that arises on shorter length scales, could be a critical experimentally observable hint at the nature of the strange metal. Moreover, quantum critical and ballistic current profiles should be clearly distinguishable in future experiments, and may give a key clue into the origin of $T$-linear resistivity: quasiparticle [41, 42] or not. Even without a high resolution image, additional measurements can shed further light into the transport physics. For example, by studying the width $w_x$ dependence of the conductance through the constriction, one can clearly distinguish between all 4 transport regimes, as summarized in Tab.1. Alternatively, we could measure vortices in a strip geometry (see the appendices). Whatever the geometry, by keeping the device *fixed* but changing the temperature $T$, one can in principle image at a low temperature $T$ where $w_x \ll \ell_{Pl}$, and a high temperature where $w_x \gg \ell_{Pl}$.

By comparing the results of imaging experiments, which reveal a fundamental scattering length $\ell$, to prior measurements of scattering times $\tau$ (e.g. [5,6]), we can extract an effective velocity scale $v = \ell/\tau$; whether or not this is the Fermi velocity or something different may be an important clue to the microscopic nature of the strange metal, and the role of electron-phonon interactions [43, 44]. In particular, if electron-phonon scattering within a Boltzmann framework captures $T$-linear resistivity, we predict an ohmic-to-ballistic crossover as constriction size approaches $\ell_{Pl}$, in contrast to the quantum critical crossover in Fig. 3.

Experimental challenges we anticipate include accurate imaging on the Planckian length scales (which can approach 10 nm or even smaller). While such magnetometers have not yet been developed to operate at the requisite temperatures to image strange metal, important progress is underway [24]. More importantly, our simple constriction geometry may be challenging to etch at 10 nm scales. Alternative approaches could use current noise [45] or the intrinsic disorder of a device to generate spatially or temporally inhomogeneous images which can be interpreted using extensions of our theoretical framework.

## Acknowledgements

We thank Sean Hartnoll for helpful comments. We especially thank the authors of [22] for their data. This work was supported in part by the Alfred P. Sloan Foundation through Grant FG-2020-13795, and through the Gordon and Betty Moore Foundation's EPiQS Initiative via Grant GBMF10279.

# A   Comments on our algorithm for calculating current flow

As been argued in the main text, the nonlocal responses due to the geometry are encoded in the effective electric field $\tilde{E}_i$. The challenging problem is to provide an actual prediction for $\tilde{E}_i$, such that we may explicit evaluate the integral in Eqn. (1) and predict a flow pattern that can be observed in experiment. Unfortunately, an exact microscopic model for $\tilde{E}_i$ does not exist, even within "canonical" frameworks like Boltzmann transport. For example, in Boltzmann kinetic theory, one must impose an infinite number of boundary conditions describing how incident particles at all momenta reflect off of the boundary. In practice, models usually assume some simple scattering mechanism at the boundary, with at most a few fitting parameters: e.g., all particles scatter off the wall at a random angle. In a similar spirit, we rely on a qualitative method first used in [25], and later [22], by solving Eqn. (1), Eqn. (2) and Eqn. (3) consistently.

Our approach only requires a computation of $\sigma_{ij}(x-x')$, which can be done *in the homogeneous theory*, together with a specification of the regions where current cannot flow. Nevertheless, we are able to calculate highly inhomogeneous flow patterns $J_i(x)$. This is possible because Eqn. (1) can be thought of as a generalization of the integral form of a standard transport equation, such as Ohm's Law. In ohmic transport, the homogeneous equation $\nabla^2\phi = 0$ can lead to inhomogeneous flows $J_i = -\sigma\nabla_i\phi$ when $\phi$ obeys non-trivial boundary conditions. For us, the non-trivial boundary conditions are imposed by Eqn. (3), which lead to $J_i = 0$ in region I. $\sigma_{ij}(x-x')$ in Eqn. (1) can loosely be understood as an analogue of the *current* Green's function $\sigma_{ij}^{\text{ohmic}} \propto \delta_{ij} - \partial_i\partial_j(\nabla^2)^{-1}$ suitable for ohmic transport: $\sigma_{ij}^{\text{ohmic}}$ encodes the fact that $J \propto E$ for transverse flows, while $\partial_i J_i = 0$. Hence, the fact that the Green's function is homogeneous does not mean it cannot generate flows through inhomogeneous regions via our algorithm. In general, $\sigma_{ij}(x-x')$ qualitatively differs from an ohmic theory, and hence $J_i(x)$ differs qualitatively for different transport regimes (e.g. ohmic vs. quantum critical).

In experiments, it is routine and straightforward to check the validity of the linear response assumption when generating images of current flow [22], so we limit ourselves to this regime. We will not calculate higher order corrections to $J_i(x)$ that arise from the accumulation of charge near region I, or any other effects second order or higher in $E_i$.

Note that our procedure does *not* generate the unique solution to Eqn. (1) [31]: the reason is that, as noted above, our ansatz Eqn. (3) amounts to a specific choice of boundary conditions. This is analogous to a well-known situation in fluid mechanics, where one may solve for the fluid flow around an object using either "no stress" (Neumann) or "no slip" (Dirichlet) boundary conditions. In, for example, the recent studies on electron hydrodynamics [22, 46], typically "no slip" assumptions are made in order to compare theory with experiment. This can be justified for a few reasons: "no slip" boundary conditions seem most compatible with data; boundary conditions interpolating between the two options typically do not lead to flows that differ qualitatively from the "no slip" flow; and, a choice simply needs to be made before experiment can be compared to a model. We anticipate that, as in this hydrodynamic problem, our simple choice of boundary conditions is likely to not have any finely-tuned parameters (since the $b \to 0$ limit simply enforces $J_i = 0$ in region I), and will thus likely model actual experiments reasonably well. Indeed, we will later compare our approach to experimental data, and find good agreement.

One check we can make is that our particular regulated $b \to 0$ limit does not change our prediction, relative to any other similar regulatory scheme. Physically, we interpret $\tilde{E}_i$ as the electric fields generated by any "space charges" which accumulate on the walls in such a way as to block any current from flowing through the constriction walls (i.e. in region I). Consider deforming $\tilde{E}_i \to \tilde{E}_i - \partial_i\tilde{\phi}$, where $\tilde{\phi}$ is the electric potential arising from some other distribution of space charges. Assuming $\tilde{\phi}$ has compact support inside region $I$, then we can extend the function $\tilde{\phi}$ to the entire plane (containing both I and O). The Fourier transform of

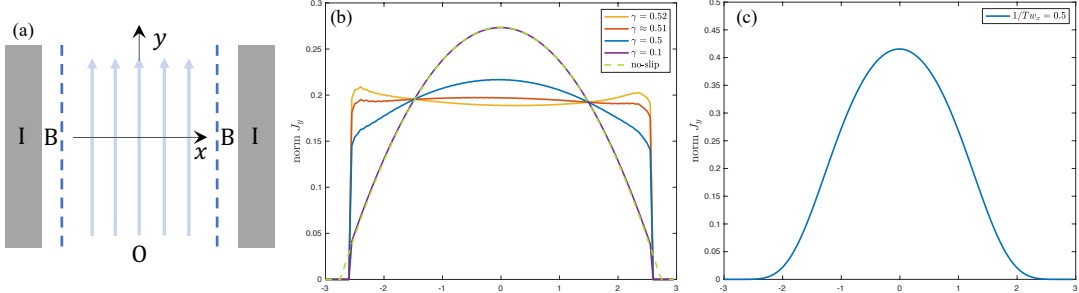

Figure 5: Current flow through a channel geometry with mixed boundary conditions. (a): we divide the computational domain into three regions: region I represents the "hard wall" that currents cannot flow ($J_i(\boldsymbol{x} \in I) = 0$); region B supports the mixed boundary conditions outside region I; region O denotes the rest domains outside region I. (b): viscous flow pattern along the line $y = 0$ in region O. Exactly at $\gamma = \gamma^* \approx 0.51$, the current profile becomes flat corresponding to the no-stress boundary condition. Above $\gamma^*$, the current profile will quickly diverge whenever $\gamma \gtrsim 0.56$; while below $\gamma^*$, the distribution evolves from flat to parabolic pattern. The dashed line is calculated with the no-slip boundary condition introduced in the main text, i.e. in the absence of region B, and it coincides with the results of $\gamma \to 0$. The difference between the dashed and solid lines in their tails is due to the finite width of region B. (c): the quantum critical flow pattern remains fixed under the mixed boundary condition.

this function is $k_i \tilde{\phi}(k)$ with $\tilde{\phi}(k)$ regular. Upon inserting such a function into Eqn. (1) we find that, upon using Eqn. (4), $\tilde{\phi}$ does not affect the current either in region I or in region O. Thus, any straightforward modification of our computational algorithm by simply changing the form of regulator $b$ (e.g. $b \to b(x)$) will not change our prediction.

## B  Adding an additional boundary layer in a channel

It is possible to take into account of other boundary conditions within our framework by assuming that $\tilde{E}_i$ exists in an additional boundary layer (which we call B). As an illustrative example, let us consider a current flowing through a channel geometry, as depicted in Figure 5. The channel extends from $x = -x_0$ to $x = +x_0$. We would like to impose the following boundary condition: [47]

$$\partial_x J_y|_{|x|=x_0} = \lambda^{-1} J_y|_{|x|=x_0}, \tag{B1}$$

where the slip length $\lambda$ allows to interpolate between the no-slip ($\lambda = 0$) and no-stress ($\lambda \to \infty$) boundary conditions, familiar from hydrodynamics. In the main text, our boundary conditions correspond to $\lambda = 0$.

In order to numerically study $\lambda > 0$, we generalize the ansatz Eqn. (3) to include a non-vanishing $\tilde{E}_i$ outside region I but within region B. Specifically, we require the current to stop in region I again: $J_i(\boldsymbol{x} \in I) = 0$, and the normal derivatives of parallel currents to vanish in region B: $\partial_x J_y(\boldsymbol{x} \in B) = 0$ (we will see how to obtain $\lambda < \infty$ below). We then solve Eqn. (1), Eqn. (2) and (B1) self-consistently according to the following scheme:

1.  Prepare the initial value for $\tilde{E}_i(\boldsymbol{x} \in I)$ according to Eqn. (3).

2. Compute $\tilde{E}_i(\boldsymbol{x} \in B)$ from the no-stress boundary condition $\partial_x J_y(\boldsymbol{x} \in B) = 0$,

$$\tilde{E}_i(\boldsymbol{x} \in B) = -\left[\partial_x \sigma_{yi}(\boldsymbol{x} \in B, \boldsymbol{x}' \in B)\right]^{-1} \circ \partial_x \sigma_{yj}(\boldsymbol{x}' \in B, \boldsymbol{x}'' \in I) \circ \tilde{E}_j(\boldsymbol{x}'' \in I), \quad (B2)$$

where $\circ$ denotes the convolution: $A(\boldsymbol{x}, \boldsymbol{y}) \circ B(\boldsymbol{y}, \boldsymbol{z}) = \int \mathrm{d}\boldsymbol{y} A(\boldsymbol{x}, \boldsymbol{y}) B(\boldsymbol{y}, \boldsymbol{z})$.

3. Solve $\tilde{E}_i(\boldsymbol{x} \in I)$ under the constraint $J_i(\boldsymbol{x} \in I) = 0$ in the presence of nonzero $\tilde{E}_i(\boldsymbol{x} \in B)$,

$$\tilde{E}_i(\boldsymbol{x} \in I) = -\left[\sigma_{ij}(\boldsymbol{x} \in I, \boldsymbol{x}' \in I)\right]^{-1} \circ \left(J_j^{(0)} + \gamma \sigma_{jk}(\boldsymbol{x}' \in I, \boldsymbol{x}'' \in B) \circ \tilde{E}_k(\boldsymbol{x}'' \in B)\right), \tag{B3}$$

where $0 < \gamma < 1$ is a parameter to control the step size, and $J_j^{(0)}$ is the constant current generated by external fields.

4. Repeat with step 2 and 3 until convergence is reached. The final current profile in region O is given by

$$J_i(\boldsymbol{x} \in O) = J_i^{(0)} + \sigma_{ij}(\boldsymbol{x} \in O, \boldsymbol{x}' \in I) \circ \tilde{E}_i(\boldsymbol{x}' \in I) + \gamma \sigma_{ij}(\boldsymbol{x} \in O, \boldsymbol{x}' \in B) \circ \tilde{E}_i(\boldsymbol{x}' \in B). \tag{B4}$$

Note that the resulting current distribution does not directly satisfy the no-stress boundary condition due to the presence of $\gamma < 1$, even if we have enforced it in step 2. However, the current profile will, in general, not be close to zero in region B, as it would in the algorithm of the main text. For this reason, we believe this simple method allows us to qualitatively capture the physics of a finite slip length $\lambda$.

In viscous fluids, we expect a parabolic current profile for no-slip boundary conditions, while a flat profile for no-stress boundary conditions [30]. In our simulation Fig. 5(a), we find the flat profile exists only with a fine-tuned step size $\gamma = \gamma^*$. By varying the step size below $\gamma^*$, the current distributions interpolate between the flat and parabolic limits. For $\gamma > \gamma^*$, our algorithm will quickly diverge; still, we find a narrow window for concave profiles to exist (see Fig. 5(b)).

In the quantum critical case, the current profile remains essentially fixed with the mixed boundary condition (Fig. 5(c)). There is no fine-tuned step size, and for all $0 \leq \gamma \leq 1$, the algorithm converges. This is heuristically understood as follows: the quantum critical flow pattern already satisfies the mixed boundary condition even derived from the no-slip boundary condition (see Appendix D).

We plan to describe more systematically the question of generalizing boundary conditions in more complicated geometries, such as the constriction, in a future paper. The primary lesson from this first example is two-fold: firstly, the algorithm described in the main text is flexible and can be generalized, and secondly, the no-slip boundary condition tends to capture more "universal" features than a no-stress-like boundary condition. Note that in Fig. 5(b), the boundary condition with a flat current profile is very finely-tuned; for any smaller value of $\gamma$, the current profile is peaked at the center of the panel, with a roughly parabolic profile between the channel walls and the center.

## C  The gravity background and conductivities

We now provide setups of the holographic correspondence. We consider the Einstein-Maxwell theory in $d = 2$ boundary spatial dimensions (or 4 bulk spacetime dimensions) [20]

$$S = \int \mathrm{d}^4 x \sqrt{-g} \left(\frac{1}{2\kappa^2}\left(R + \frac{6}{L^2}\right) - \frac{1}{4e^2} F^2\right). \tag{C1}$$

The static and isotropic metric solving the equation of motions is the AdS$_4$-RN geometry [48],

$$ds^2 = \frac{L^2}{r^2}\left(-f(r)dt^2 + \frac{dr^2}{f(r)} + dx^2 + dy^2\right),$$ (C2)

where the emblackening factor is

$$f(r) = 1 - \left(\frac{1}{r_+^3} + \frac{\mu^2}{r_+\gamma^2}\right)r^3 + \frac{\mu^2}{r_+^2\gamma^2}r^4.$$ (C3)

The spacetime has a horizon at $r = r_+$ with Hawking temperature

$$T = \frac{1}{4\pi r_+}\left(3 - \frac{\mu^2 r_+^2}{\gamma^2}\right).$$ (C4)

We have grouped the coefficients into $\gamma^2 = 2e^2L^2/\kappa^2$ representing the relative strength of the couplings. The profile of the U(1) gauge field is $A_t = p(r) = \mu - e^2\rho r$ with $\mu = e^2\rho r_+$. To further facilitate our calculations, we define the dimensionless parameters

$$Q \equiv \frac{\mu r_+}{\gamma}, \quad u \equiv \frac{r}{r_+}, \quad w \equiv \omega r_+, \quad q \equiv k r_+.$$ (C5)

All radial derivatives below, denoted with primes, refer to $\partial/\partial u$.

To calculate $\sigma(k)$ holographically, we must find the equations of motion of the Einstein-Maxwell theory, and subsequently linearize them about (C2). Ultimately, this will lead to second-order ordinary differential equations corresponding to fluctuations in the bulk fields:

$$\delta A_\mu = a_\mu(r)e^{-i\omega t + ikx},$$ (C6a)

$$\delta g^\mu{}_\nu = h^\mu{}_\nu(r)e^{-i\omega t + ikx}.$$ (C6b)

The direction of momentum is chosen as $\boldsymbol{k} = k\hat{x}$ without loss of generality (the background is rotation invariant in $x$ and $y$). Using parity ($y \to -y$) symmetry, we can divide the perturbation modes into transverse modes (odd under parity): $a_y, h^t{}_y, h^x{}_y$ and longitudinal modes (even under parity): $a_t, a_x, h^t{}_t, h^t{}_x, h^x{}_x, h^y{}_y$. Only the odd modes contribute to $\sigma(k)$. Note that we are working in the gauge $a_r = h_{r\mu} = 0$.

# D Heuristic argument for the sinusoidal quantum critical current profile

Here we present a more quantitative argument for the flow pattern in a quantum critical regime. For simplicity, let us begin by calculating flow patterns in a channel, in which current is restricted to flow in the region $|x| \le w_x/2$; we will then argue that our conclusions do not qualitatively change in a constriction.

Imagining putting the flow onto a periodic grid with $x \sim x + w_x$ identified, and restricting to the line $y = 0$, [49] showed that given the ansatz

$$\tilde{E}_y(x) = -a\sum_{p\in\mathbb{Z}} J_y(x_p)\delta(x - x_p),$$ (D1)

with $x_p = \pm w_x p/2$ and $a \sim b^{-1} \to \infty$ (cf. Eqn. (3)), the distribution of current will be given by

$$J_y(x, y = 0) = J_0\left(1 - \frac{\sum_{n=1}^\infty c_n \cos(k_n(x + w_x/2))}{\sum_{n=1}^\infty c_n}\right),$$ (D2)

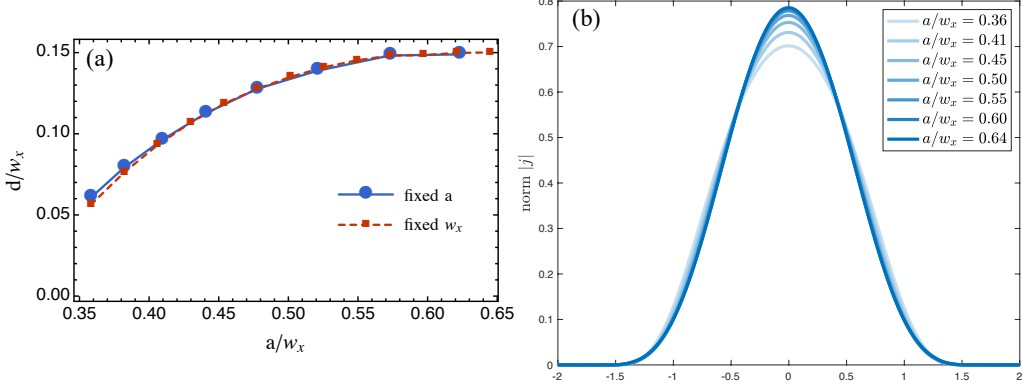

Figure 6: Numerical test of the heuristic argument for the sinusoidal current profile $J_y(x)$ with quantum critical conductivity $\sigma(k) = \exp(-ak)$. (a): the best fitting distance $d$ (D4) versus the exponent $a$ and the width $w_x$. We obtain the blue curve by a fixed exponent $a = 1.43\,\mu$m, the red curve by a fixed width $w_x = 3\,\mu$m. Upon rescaling $1/w_x$ (for blue) as well as $a$ (for red) to $a/w_x$, we find that the curves coincide. In the transient regime $a/w_x \lesssim 0.5$, $d$ increases with respect to $a/w_x$. When $a/w_x \gtrsim 0.5$, however, they saturate to $d \approx w_x/7$, indicating a non-changing profile. This is supported by plot (b), which demonstrates that the current profiles collapse once $a/w_x \gtrsim 0.5$: the spatial profiles plotted here correspond to data taken at every other square (red) plot point in (a).

where $c_n = \sigma(k_n = 2\pi n/w_x)$ in a channel, and $J_0$ is the total current. Observe that if we had ohmic flow, where $c_n = $ constant, we would have $J_y = J_0$ a uniform current distribution. If we instead take $c_n \propto 1/n^2$, we would find precisely the quadratic flow profile $J_y \propto (w_x/2)^2 - x^2$ from Poiseuille flow [30] in a viscous fluid. We saw that, when $\sigma(k)$ is given by Eqn. (8) in the quantum critical regime, $c_n \approx e^{-\alpha^* n}$ for a constant $\alpha^* \propto \ell_{\text{Pl}}/w_x$. Hence (D2) is dominated only by the $n = 1$ mode $J_y \propto 1 + \cos(2\pi x/w_x)$ showing a sinusoidal profile (up to a constant shift). This agrees with our expectations that the lowest Fourier modes dominate $J_y(x)$, stated in the main text.

Now that we are confident this method captures flow patterns in a channel, we may ask what happens when we change the geometry to a constriction geometry. In this case, we know the form of the current profiles in ohmic ($J_y \propto [(w_x/2)^2 - x^2]^{-1/2}$) and viscous ($J_y \propto [(w_x/2)^2 - x^2]^{+1/2}$) regimes [31]. We find that these imply (asymptotically) $c_n \propto n^{-1/2}$ and $c_n \propto n^{-3/2}$ respectively. A crude formula that relates the two would be:

$$c_n^{\text{constriction}} \propto \sqrt{c_n^{\text{channel}}/n}. \tag{D3}$$

We certainly do not claim that this is a generic result. But, at least taking the trend suggested by (D3) seriously, we expect that since, in the quantum critical regime, $c_n^{\text{channel}} \propto e^{-\alpha^* n}$, (D3) implies that $c_n^{\text{constriction}} \propto e^{-\alpha^* n/2}/\sqrt{n}$. This is qualitatively the same current flow pattern, and is still dominated by the $n = 1$ mode.

The argument above is not mathematically rigorous: in particular, it does not capture boundary effects in the quantum critical regime. To confirm our expectations that again $J_y(x)$ is dominated by long wavelength Fourier modes, we have numerically studied the current flow profiles, where we find that, for a quantum critical flow, the boundary is effectively pushed in by a distance $d$. At sufficiently small $w_x/\ell_{\text{Pl}}$, $d$ saturates to a constant $d \approx w_x/7$. More specifically, we choose manually $\sigma(k) = e^{-ak}$ with exponent $a \propto \ell_{\text{Pl}}$ in our numerics. Solving for the current flow pattern through the constriction, we fit the current distribution $J_y(x)$ with

a modified sinusoidal function

$$J_y = A\left(1 + \cos\left(\frac{2\pi x}{w_x - d}\right)\right), \tag{D4}$$

where $A$ and $d$ are two fitting parameters. We set a current cutoff $10^{-3}$ to discard regions that is originally outside the constriction but now carries nearly zero current. We consider two cases: one with a fixed exponent $a$, and one with a fixed width $w_x$. As shown in Fig. 6(a), the curves from the two cases collapse onto a single one upon rescaling $a, 1/w_x \to a/w_x$. The distance $d$ increases with respect to $a/w_x$ in the transient regime when $0.3 \lesssim a/w_x \lesssim 0.5$ (for $a/w_x \ll 1$ the flow appears ohmic). When $a/w_x \gtrsim 0.5$, however, the flow enters a strongly quantum critical regime with $d \approx w_x/7$, and we find that consistent with our expectations – the current profile asymptotes to a universal curve dominated by approximately a single "sine wave": see Fig. 6(b).

# E   Critical transport at zero density

In this section, we will calculate the spectral weight $\sigma(k)$ both numerically and analytically. At zero density, the perturbation of the gauge field will decouple from gravitational modes, and the equation of motion for the transverse mode $a_y$ becomes

$$\left(f(u)a_y'\right)' + \left(\frac{w^2}{f(u)} - q^2\right)a_y = 0. \tag{E1}$$

We impose the infalling boundary condition at horizon $u = 1$, which implies

$$a_y(u \to 1) = (1-u)^{-iw/3}F(u), \tag{E2}$$

where $F(u)$ is a regular function about $u = 1$. Plugging this into (E1), and taking the $u \to 1$ limit, we find that

$$F(u) \approx 1 - (1-u)\left(\frac{iw}{3} - \frac{q^2}{3-2iw}\right). \tag{E3}$$

The overall constant of proportionaliy in $F(u)$ is not important in calculating $\sigma(k)$ [20]. Now we solve (E1) numerically in the domain $u \in [0,1]$, using the boundary condition (E3) at horizon. The resulting spectral weight, calculated via Eqn. (5) in the main text, is shown in Fig. 7(a).

## A. Small $q$

To better understand the behavior of $\sigma(q)$ at finite $q$, we first use a perturbative approach to study the effects of small $q$ ($q \ll 1$, or $k \ll T$ in more physical units), following the Wronskian construction in [50]. The transverse modes $a_y$ can be expanded at small $\omega$ as

$$a_y(q,u) = a_{y,0}(q,u) + iw a_{y,0}(q,1)^2 a_{y,1}(q,u), \tag{E4}$$

where $a_{y,1}$ is the Wronskian partner of the regular solution $a_{y,0}$. We now take the normalization $a_{y,0}(q, u \to 0) = 1$. The conductivity therefore becomes

$$\text{Re}\,\sigma(q) = \frac{1}{e^2}a_{y,0}(q,1)^2. \tag{E5}$$

Next, we expand $a_{y,0}$ with respect to $q \ll 1$:

$$a_{y,0}(q,u) \approx a_{y,0}^{(0)}(u) + q^2 a_{y,0}^{(1)}(u) + O(q^4). \tag{E6}$$

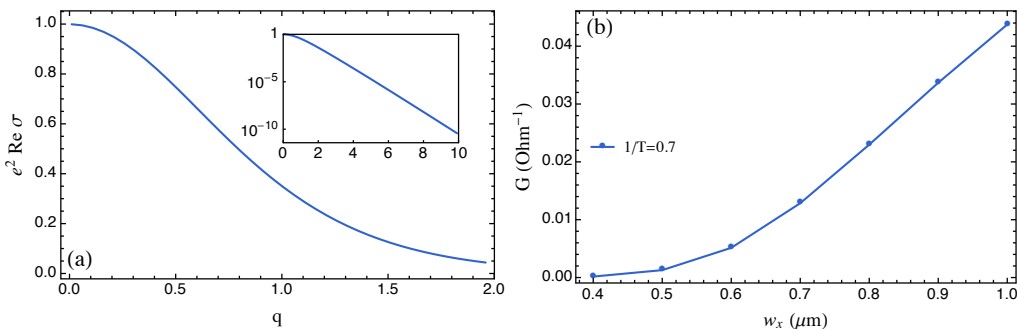

Figure 7: Spectral weight and conductance at zero density. (a): momentum dependence of spectral weight of current. Inset shows the exponential decay of the spectral weight with respect to $q$. (b): Numerical result matches the heuristic argument $G \sim \sigma(k = 1/w_x)$ at $w_x < 1/T$ (see S9

The reason why $q^2$ is the leading order correction is that there is no Linear-in-$q$ terms in the equation of motion. The above modes then satisfy

$$\left(f(u)a_{y,0}^{(0)\prime}(u)\right)' = 0, \tag{E7a}$$

$$\left(f(u)a_{y,0}^{(1)\prime}(u)\right)' = a_{y,0}^{(0)}(u). \tag{E7b}$$

Solving them with regular solutions, we obtain

$$a_{y,0}(q,u) \approx 1 - \frac{2}{\sqrt{3}}\left[\arctan\left(\frac{1+2u}{\sqrt{3}}\right) - \frac{\pi}{6}\right]q^2 + O(q^4), \tag{E8}$$

where integral constants are chosen to satisfy the normalization condition. Then, plugging it into (E5), we arrive at

$$\text{Re }\sigma(q) \approx \frac{1}{e^2}\left(1 - \frac{2\pi}{3\sqrt{3}}q^2 + O(q^4)\right), \tag{E9}$$

where the leading order correction is $O(q^2)$.

**B. Large $q$**

Now consider the regime where $q \gg 1$ ($k \gg T$ in more physical units). If the system were at $T = 0$ ($q \to \infty$ limit here), the solution would be exactly

$$a_{y,0}(q) \approx e^{-qu}. \tag{E10}$$

The near horizon solution should therefore be exponentially suppressed. Following the Wronskian argument above, we may conclude that $\text{Re }\sigma(q) = a_{y,0}(q,1)^2 \propto e^{-2q}$.

This idea becomes more concrete when taking the WKB limit of the equation of motion. Specifically, we write (E1) as a "Schrödinger-like" equation

$$-\partial_{u_*}^2 a_y + V(u)a_y = w^2 a_y, \tag{E11}$$

where $\partial_{u_*} = f(u)\partial_u$ and $V(u) = q^2 f(u)$. Following the standard manipulation [20], we arrive at

$$\text{Im }G_{J_y J_y}^{R}(q,w) \propto e^{-2qu_o} \times \exp\left\{-2\int_0^{u_o} ds\left(\sqrt{\frac{q^2}{f(s)} - \frac{w^2}{f(s)^2}} - q\right)\right\}, \tag{E12}$$

where $u_o$ is the turning point at which $V(u_o) = w^2$ and should be approximated as $u_o \approx 1$ with large $q$ and small $w$. The exponential decay thus emerges from the "Boltzmann" weight of the WKB limit, from which we extract the critical $q$ as

$$q^* = \frac{1}{2\int_0^1 ds/\sqrt{f(s)}} = \frac{1}{2\,_2F_1(\frac{1}{3}, \frac{1}{2}, \frac{4}{3}, 1)} \approx 0.36. \tag{E13}$$

# F    Critical transport at finite density

Now we perturb the bulk theory by a finite chemical potential. The excitations of bulk gravitational modes now mix with the gauge field fluctuations, resulting in coupled Einstein-Maxwell equation of motions. Fortunately, the linearized equation of motions for the transverse modes can be decoupled by the master fields [51]

$$\Phi_\pm(u) = -\frac{\mu}{u}\frac{qf(u)}{w^2 - q^2 f(u)}(qh_t^{y\prime} + wh_y^{x\prime}) - \left(\frac{4Q^2q^2f(u)}{w^2 - q^2f(u)}u + 2g_\pm(q)\right)a_y, \tag{F1}$$

where

$$g_\pm(q) = \frac{3}{4}\left(1 + Q^2\right) \pm \sqrt{\frac{9}{16}\left(1 + Q^2\right)^2 + q^2Q^2}, \tag{F2}$$

with the constraint $0 < Q^2 < 3$. The decoupled equation of motions are

$$\left(f(u)\Phi_\pm'\right)' + \left(-\frac{f'(u)}{u} + \frac{w^2 - q^2f(u)}{f(u)} - 2g_\pm(q)u\right)\Phi_\pm = 0. \tag{F3}$$

Similar to E, we write the ansatz as

$$\Phi_\pm(u \to 1) = (1 - u)^{-iw/(3 - Q^2)}F_\pm(u), \tag{F4}$$

with

$$F_\pm(u) \approx 1 - (1 - u)\frac{3 - Q^2 - 2g_\pm - q^2 + iw(3 - 3Q^2)/(3 - Q^2) + w^2(6 - 2Q^2)(3 - 3Q^2)/(3 - Q^2)^3}{3 - 2iw - Q^2}, \tag{F5}$$

being found by taking $u \to 0$ in equation of motions. In terms of the master fields, the spectral weight becomes [51]

$$\begin{aligned}
\mathrm{Re}\,\sigma(q) &= \mathrm{Re}\,\sigma_+(q) + \mathrm{Re}\,\sigma_-(q) \\
&= \chi \lim_{\omega \to 0}\frac{1}{\omega}\left(\frac{g_+(q)}{g_+(q) - g_-(q)}\mathrm{Im}\,G_{\Phi_+\Phi_+}^R(q, w) - \frac{g_-(q)}{g_+(q) - g_-(q)}\mathrm{Im}\,G_{\Phi_-\Phi_-}^R(q, w)\right),
\end{aligned} \tag{F6}$$

and their dependence on $q$ and $Q$ is plotted in Fig. 8. In Fig. 8(a), we find that the spectral weight $\sigma_+$ associated with the $\Phi_+$ mode reduces to our prior results exactly at $Q = 0$, indicating that the $\sigma_+$ contribution to spectral weight contains the incoherent conductivity (current flow which does not arise from momentum dynamics). Moving towards larger density, contribution of the incoherent conductivity gets smaller. To identify the critical momentum above which the $\sigma_+$ starts to drop, a similar WKB method to the above can be carried out:

$$q^*(Q) = \frac{1}{2\int_0^1 ds/\sqrt{f(s)}} = \frac{1}{2\int_0^1 ds/\sqrt{1 - (1 + Q^2)s^3 + Q^2s^4}}. \tag{F7}$$

We find a drop of $q^*$ when $Q \to \sqrt{3}$ in Fig. 8(a).

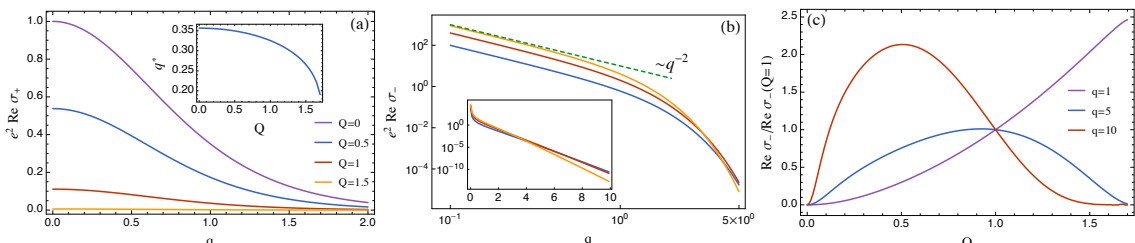

Figure 8: Spectral weight at finite desnity. (a): $\sigma_+$ against $q$ for various $Q$. Inset shows the critical $q^*$ from the WKB limit. (b): $\sigma_-$ against $q$ for $Q = 0.5, 1, 1.5$. All the curves scale as $q^{-2}$ at small $q$. Inset shows that the exponential decay at large $q$ depends on $Q$. The dependence is more clear in (c), where the normalized $\sigma_-$ becomes non-monotonic against $Q$ for large $q$.

Meanwhile, Fig. 8(b) shows a singular scaling for the spectral weight associated with the $\Phi_-$ mode at small $q$:

$$\text{Re } \sigma_- \sim \frac{1}{q^2}. \tag{F8}$$

This divergence arises from the $q^{-2}$ divergence in hydrodynamic spectral weight at finite density, arising from viscous effects. Hence, $\sigma_-$ contains the "Drude weight" from the frequency-dependent electrical conductivity. To see it more clearly, we extend the hydrodynamic result to finite momentum using the quasinormal mode result $\sigma_-(w, q) \sim (q^2 - iw)^{-1}$, and we find $\sigma_-(\omega)|_{q \to 0} \sim i/\omega + \pi\delta(\omega)$, where the delta function comes from the identity $1/(x + i\epsilon) = 1/x + i\pi\delta(x)$.

## A. Spectral weight in an extremal black hole

At large $q$, $\sigma_{\pm}$ decay exponentially, for analogous reasons to what we observed at zero density. To quantify this more directly, we may use a standard matching argument, taking advantage of an emergent IR $\text{AdS}_2 \times \mathbb{R}^2$ geometry [20]. Indeed, we note that if $T = 0$, the equation of motion (F3) for master field $\Phi_-$ reduces to

$$\left(f(u)\Phi_-'\right)' + \left(12u(1-u) + \frac{w^2}{f(u)} - q^2 - 6ug_-(q)\right)\Phi_- = 0, \tag{F9}$$

where $f(u) = 1 - 4u^3 + 3u^4$, and $g_-(q) = 1 - \sqrt{1 + q^2/3}$.

Here, we first work in the limit $k \ll T \ll 1$.

**Inner region** ($u \to 1$): let us define the new radial coordinate

$$\zeta = \frac{w}{6(1-u)}. \tag{F10}$$

Then (F9) becomes, in the near horizon limit $\zeta/w \to \infty$,

$$\partial_\zeta^2 \Phi_- + \left(1 - \frac{q^4/72}{\zeta^2}\right)\Phi_- = 0. \tag{F11}$$

This is the equation of motion for $\text{AdS}_2 \times \mathbb{R}^2$ spacetime [20].

The exact solution can be found as

$$\Phi_-(\zeta) = a_{-,I}\left(1 + \frac{i}{\zeta}\frac{q^4}{144}\right)e^{i\zeta} \approx a_{-,I}\left(\frac{i}{\zeta}\frac{q^4}{144}(1 + \ldots) + 1 + \ldots\right), \tag{F12}$$

where in the last step, we expand the solution into $\zeta \to 0$ limit for further matching argument.

**Outer region** ($u \to 0$): in the near boundary region, we can safely set $w = 0$ and $q = 0$, where later is due to the fact that the UV boundary theory is insensitive to small $k$. The (F9) now becomes

$$\left(f(u)\Phi_-'\right)' + 12u(1-u)\Phi_- = 0. \tag{F13}$$

The solution to it is given by

$$\Phi_-(u) = c_1 u + c_2 \left\{ \frac{7u-6}{6(1-u)} - \frac{u}{36}\left[23\sqrt{2}\arctan\left(\frac{1+3u}{\sqrt{2}}\right) + 20\log(1-u) - 10\log(1+2u+3u^2)\right]\right\}. \tag{F14}$$

The asymptotic behaviors of the above solution are

$$\Phi_-(u \to 0) = -c_2 + \left(c_1 + \frac{c_2}{36}(6 - 23\sqrt{2}\operatorname{arccot}(\sqrt{2}))\right)u + \dots, \tag{F15a}$$

$$\Phi_-(u \to 1) = \left(c_1 - \frac{c_2}{36}(42 + 23\sqrt{2}\arctan(2\sqrt{2}) - 10\log 6)\right)$$
$$- \left(c_1 + \frac{c_2}{108}(17 - 69\sqrt{2}\arctan(2\sqrt{2}) + 30\log 6)\right)(1-u) + \dots. \tag{F15b}$$

**Matching:** comparing the same order of $(1-u)$ and $(1-u)^0$ for $\Phi_-$ in the overlap region: $\zeta \to 0$ from inner region, and $u \to 1$ from outer region, together with (F15a), we find

$$\operatorname{Im} G^R_{\Phi_-\Phi_-} \propto \frac{w}{q^4}. \tag{F16}$$

Taking into account of the weight (F6), we thus obtain (F8).

Now, let us study the limit $T \ll k$. A black hole arises from the $AdS_2 \times \mathbb{R}^2$ spacetime with the horizon located in the $AdS_2$ coordinate $\zeta$ at $\zeta_+ \propto 1/T$. The imaginary part of the retarded Green's function can be found as [20]

$$\operatorname{Im} G^R_{\Phi_-\Phi_-} \sim (T/\mu)^{2\nu_q - 1}, \tag{F17}$$

where

$$\nu_q = \frac{1}{2}\sqrt{1 + 4\left(q^2/6 + 1 - \sqrt{1+q^2/3}\right)} \tag{F18}$$

is determined through (F9) in the IR scaling region. We find that

$$\operatorname{Im} G^R_{\Phi_-\Phi_-}(q \ll 1) \sim 1,$$
$$\operatorname{Im} G^R_{\Phi_-\Phi_-}(q \gg 1) \sim (T/\mu)^{\sqrt{\frac{2}{3}}q}, \tag{F19}$$

thus, at $q \gg 1$, $\sigma_-$ will go to zero with $\mu \to \infty$ ($Q \to \sqrt{3}$). In other words, the exponential decay coefficient now depends on both $\mu$ and $T$.

## B. Current distributions

To sketch out the phase diagram spanned by $\mu$ and $T$, we plot the curvature ($\partial_x^2|j|$) at the center of the constriction in Fig. 9(a). Three regimes are clearly visible in the plot: (I) ohmic transport driven by incoherent conductivity for $\mu \ll T$ (approximately zero density), and $w_x \gtrsim 1/T$; (II) viscous transport when $\mu \gtrsim T$, and $w_x \gtrsim 1/\max(T,\mu)$; (III) quantum critical transport when $w_x \lesssim 1/\max(T,\mu)$. At large $\mu$, in this holographic model, we observed that it is *not* the Planckian length scale that governs the crossover to "quantum critical" current profiles; this appears related to the existence of hydrodynamic sound modes with wavelength $1/\mu$ [52]. This particular feature of our model may not generalize to other models of quantum critical dynamics. In Fig. 9(b), we showed the 2D current flow pattern as a supplementary to the current distributions at $y = 0$ in the main text.

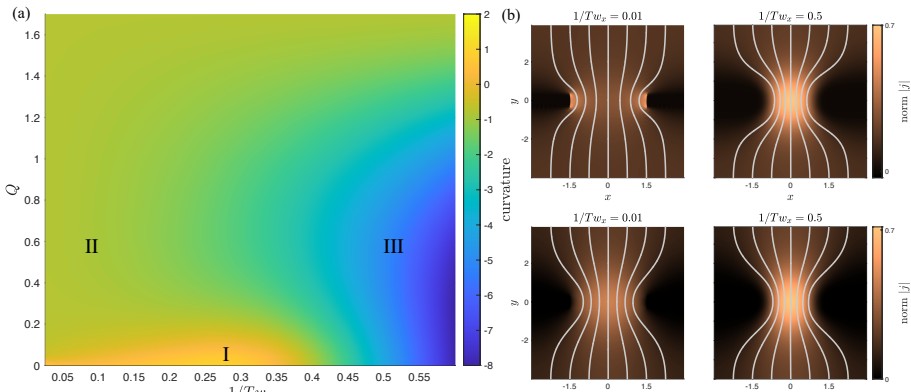

Figure 9: (a): curvature of the current profile at the constriction center, given by $d^2|j|(y = x = 0)/dx^2$, against $T$ and $Q$. Here $Q$ reparameterizes $\mu$ via $Q \propto \mu r_+$, $4\pi T r_+ = 3 - \mu^2 r_+^2$. There are three different regimes: (I) the Ohmic regime, (II) the viscous regime, and (III) the quantum crtical regime. (b): 2D current distribution and streamlines for $1/Tw_x = 0.01$ and $0.5$ with the same slit width as in the main text. The top line is at zero density ($Q = 0$), while the bottom line is at finite density ($Q = 0.5$).

## G Details of comparison to experiments

We now detail how we analyzed the experimental data from [22], which imaged current flow through a constriction in monolayer graphene near the charge neutrality point at temperatures 128 K and 297 K. (We leave further details on the experimental techniques to [22]). Given a raw image of two dimensional data for the currents $j_{x,y}(x, y)$, with the $x$ and $y$ coordinates aligned as in the main text, we focus on analyzing the magnitude of currents $|j|$ along a fixed $y = y_c$ line. We symmetrize the data about the point $x = x_c$. Here, the points $x_c, y_c$ represent the central points that we must determine.

To optimize $x_c, y_c$ as well as the free parameter $C$ (defined in the main text), we proceed as follows. First, we take $C \approx 0.2$, as reported in [12]. We then fix $x_c$ and $y_c$ by minimizing the root mean square error (RMSE) on the resulting fits. Once $x_c$ and $y_c$ are determined, we then find the value of $C$ which minimizes RMSE between our theory and experiment. Because the scanning resolutions of the experimental magnetometry were 0.1441 $\mu$m and 0.1478 $\mu$m for 297 K and 128 K, respectively, we applied a Gaussian filter to our theoretical simulation to mimic the smearing of current, as imaged by the limited-resolution magnetometer [22]. The results of our analysis were highlighted in the main text.

We subsequently fitted this experimental dataset with simple kinetic theory model of transport, which assumes a well-defined Fermi surface. While charge neutral graphene does not have a Fermi surface, these models have been used previously [23] to fit imaging data near charge neutrality. Using the Boltzmann model of [22], which takes in *two* input parameters $\ell_{ee}$ (momentum-conserving scattering length) and $\ell_{mr}$ (momentum-relaxing scattering length):

$$\sigma(k) = \frac{2\ell_{ee}\ell_{mr}}{2\ell_{ee} - \ell_{mr} + \ell_{mr}\sqrt{1 + k^2\ell_{ee}^2}}, \tag{G1}$$

we find that the optimal fits have $\ell_{ee} \approx 80$ nm at both temperatures, while $\ell_{mr} \approx 1300$ nm at 128 K and 150 nm at 297 K (Fig. 10(b)); RMSEs are similar for both the holographic and the kinetic models. We emphasize that these parameters are not very tightly constrained by the kinetic model [22]: the quality of fit in this context is sensitive to $\sqrt{\ell_{ee}\ell_{mr}}$ [30], which appears optimized close to $\ell_{qc}$ in our holographic fit. As this length scale signals the breakdown

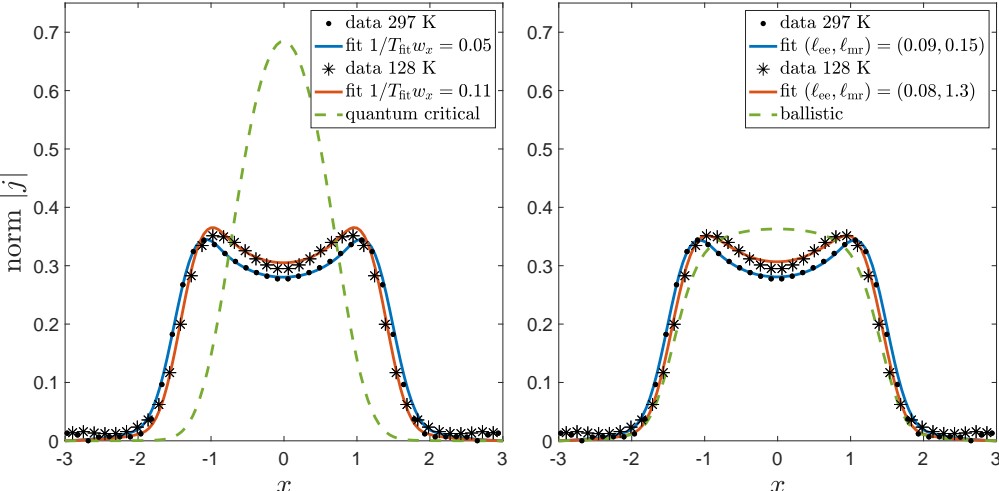

Figure 10: A comparison between holographic (left) and kinetic (right) predictions. The holographic fitting has already been presented in the main text. The solid lines indicate the best fit; while the green curves predicts for a constriction of width 600 nm at experimental temperature $T = 128$ K.

of hydrodynamics, we believe that the holographic model is superior to the single Fermi surface model.

Another crosscheck on the nature of transport can be done by looking into the structure of vortices in a strip geometry. As shown in Appendix J, the quantum critical flow develops a multi-vortex structure, while in a kinetic model with $\sqrt{\ell_{ee}\ell_{mr}} \sim 1/T$, the number of vortices is limited (one for ballistic flow, while two for viscous flow [53]).

## H  Free Dirac fermion

Here, we apply the field theory to study non-interacting Dirac fermions in $(2 + 1)$-D. The Euclidean correlator is [20]

$$\langle J_\mu(K)J_\nu(-K)\rangle = T\sum_{p_n}\int\frac{\mathrm{d}^2p}{(2\pi)^2}\frac{\mathrm{tr}[\gamma_\mu\gamma_\lambda P_\lambda\gamma_\nu\gamma_\delta(K_\delta - P_\delta)]}{P^2(K-P)^2}$$

$$= T\sum_{p_n}\int\frac{\mathrm{d}^2p}{(2\pi)^2}\left[\frac{2\delta_{\mu\nu}}{P^2} + \frac{-K^2\delta_{\mu\nu} + 2K_\mu K_\nu - 4P_\mu P_\nu}{P^2(K-P)^2}\right], \qquad \text{(H1)}$$

where $K_\mu = (k_n, \boldsymbol{k})$ and $k_n = (2n+1)\pi T$, and in the second step we apply the change of variables $P \to K - P$ in half of the equations. The first term – it is related to the Drude weight – is always real and is not our interest here, thus we focus on the second term. After performing the Matsubara summation and analytically continuing it to real frequencies, we obtain

$$\mathrm{Im}\,G^R_{J_iJ_j}(\omega, k) = \mathrm{Im}\sum_{ss'}\int\frac{\mathrm{d}^2p}{(2\pi)^2}\frac{L_{ij}}{4E_pE_{k-p}}\frac{ss'[n_F(sE_p) + n_F(s'E_{k-p}) - 1)]}{\omega + \mathrm{i}\epsilon - sE_p - s'E_{k-p}}$$

$$= -\sum_{ss'}\int\frac{\mathrm{d}^2p}{(2\pi)^2}\frac{\pi L_{ij}}{4E_pE_{k-p}}ss'[n_F(sE_p) + n_F(s'E_{k-p}) - 1)]\delta(\omega - sE_p - s'E_{k-p}),$$

$$\text{(H2)}$$

where, by taking $\boldsymbol{k} = (k, 0)$ and $\boldsymbol{p} = p(\cos\theta, \sin\theta)$,

$$L_{xx} = \omega^2 + k^2 - 4p^2 \cos^2\theta \,, \tag{H3a}$$

$$L_{yy} = \omega^2 - k^2 - 4p^2 \sin^2\theta \,. \tag{H3b}$$

The real part of the conductivity is again given by

$$\operatorname{Re} \sigma(k) \equiv \lim_{\omega \to 0} \frac{\operatorname{Im} G^R_{J_y J_y}(\omega, k)}{\omega} \,. \tag{H4}$$

We first focus on the low temperature limit: $\omega \ll T \ll k$. The (H2) is nonzero only if $ss' = -1$ and the two cases equal to each other. The delta function reduces to

$$\delta(\omega - E_p + E_{k-p}) = \delta(p - p_0) \left| 1 - \frac{p - k\cos\theta}{E_{k-p}} \right|^{-1}, \tag{H5}$$

where

$$p_0 = \frac{\omega^2 - k^2}{2(\omega - k\cos\theta)} \,. \tag{H6}$$

Then, (H2) becomes

$$\operatorname{Im} G^R_{J_y J_y}(\omega, k)|_{ss'=-1} = \frac{1}{2\pi^2} \int_{-\arccos\frac{\omega}{k}}^{\arccos\frac{\omega}{k}} \mathrm{d}\theta \int_0^\infty \mathrm{d}p p \frac{\pi L_{yy}[n_F(E_p) - n_F(E_{k-p})]}{4 E_p E_{k-p} \left| 1 - \frac{p - k\cos\theta}{E_{k-p}} \right|} \delta(p - p_0). \tag{H7}$$

Taking the linear dispersion relation of Dirac fermions $E_p = p$ and $E_{k-p} = \sqrt{(k - p\cos\theta)^2 + p^2 \sin^2\theta}$, we find approximately

$$
\begin{aligned}
\operatorname{Re} \sigma(k) &\approx \lim_{\omega \to 0} \frac{1}{\omega} \frac{-k}{8\pi} \int_{-\pi/2}^{\pi/2} \mathrm{d}\theta \frac{1}{\cos^3\theta} \left[ \frac{1}{e^{\beta p_0} + 1} - \frac{1}{e^{\beta(p_0 - \omega)} + 1} \right] \\
&\approx \frac{k\beta}{4\pi} \int_0^{\pi/2} \mathrm{d}\theta \frac{\exp(-\beta \frac{k}{2\cos\theta})}{\cos^3\theta} \\
&= \frac{k\beta}{4\pi} K_0(k\beta/2) + \frac{1}{2\pi} K_1(k\beta/2),
\end{aligned} \tag{H8}
$$

where $K_n(z)$ is the modified Bessel function of the second kind, and it decays exponentially at large $z$. Next, the high temperature limit $\omega \ll k \ll T$ is considered in the appendix of [54], where they found

$$\operatorname{Im} G^R_{J_y J_y}(z) \approx -\operatorname{Im} \frac{T \log 2}{2\pi} \left[ \frac{z(2 - 2z^2)}{\sqrt{z^2 - 1}} + 2z^2 \right], \tag{H9}$$

where $z \equiv (\omega + i\epsilon)/k$. The real part of the conductivity thus becomes

$$\operatorname{Re} \sigma(k) \approx \frac{T \log 2}{\pi k} \,. \tag{H10}$$

We find that it has the same scaling $\sim 1/k$ as the ballistic transport with a Fermi surface.

The numerical interpolation of the conductivity between the high and low temperature limits is shown in Fig. 11(a). In Fig. 11(b) we plot the predicted current profiles for free Dirac fermions.

More generally, we emphasize that whenever there is a finite "Fermi surface", we expect to have $\sigma(k) \sim k^{-1}$ on length scales short compared to any interaction scales, but long compared

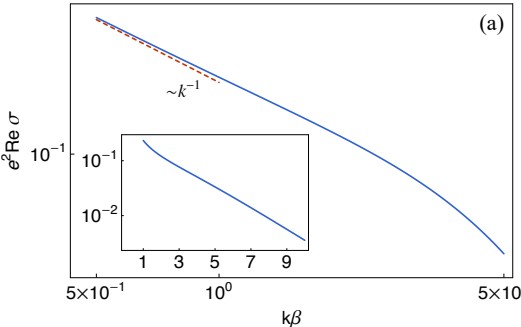
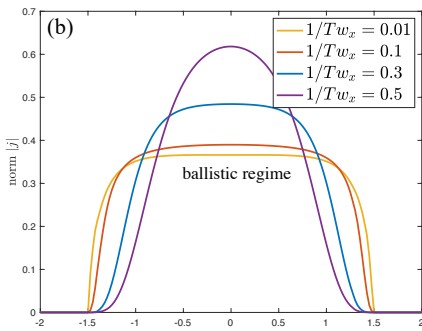

Figure 11: Conductivity and current profiles for free Dirac fermions. (a): the conductivity scales $\sim 1/k$ at $k\beta \ll 1$ while decays exponentially at $k\beta \gg 1$ as shown in the inset. (b): the predicted current profile evolves from a flat to a singly peaked distribution upon decreasing the temperature.

to the Compton wavelength (i.e. $k \ll k_{\mathrm{F}}$) [55]. In this thermal case, we have $k_{\mathrm{F}} \to T$. This scaling follows from the fact that $\sigma(k)$ becomes dominated by quasiparticles obeying $\boldsymbol{v} \cdot \boldsymbol{k} = 0$ near the Fermi surface. Schematically (see [55] for a more formal discussion), in the ballistic limit of a kinetic theory, one finds that, after ignoring the Drude weight,

$$\sigma(k) \sim \int\limits_{\mathrm{FS}} \mathrm{d}^{d-1}p \; v^2 \delta(\boldsymbol{v_p} \cdot \boldsymbol{k}) \sim k^{-1}, \tag{H11}$$

where the factor of $k^{-1}$ comes from the identity $\delta(ax) = \delta(x)a^{-1}$ and from the integral over the Fermi surface. In particular, this argument demonstrates that if electron-phonon scattering is responsible for Planckian resistivity, *and* an electronic quasiparticle is still well-defined on length scales short compared to the Planckian length, then regardless of the Fermi surface or microscopic model, we will find $\sigma(k) \sim k^{-1}$ when $k\ell_{\mathrm{Pl}} \gg 1$. This is sharply contrasted with the quantum critical case of either the free Dirac fermions or the holographic models described in the main text.

# I Conductance

As highlighted in the main text, we can use (4) to determine the response of other fields to the application of a uniform electric field (up to $\tilde{E}_i$ generated by the constriction). For practical purposes, we will focus on the choice $\mathcal{O} = n$, the charge density, in applying the generalized response equation (4). This will allow us to calculate the conductance $G \equiv I/V$ (or equivalently the resistance $R \equiv G^{-1}$) associated to the constriction; here $I$ is defined to be the total current along any fixed line $y = y_0$, and $V$ is the potential difference across the constriction, measured at large distance $y \gg w_{x,y}$ (note the value of $x$ will not be important). To obtain $V$, the potential distribution is required. Since chemical potential and voltage are thermodynamically conjugate to density, we can calculate $V$ by choosing $\mathcal{O} = n$. Indeed, charge density is related to the chemical potential through

$$n(\boldsymbol{x}) = \int \mathrm{d}^2x' \chi_{nn}(\boldsymbol{x} - \boldsymbol{x}')\mu(\boldsymbol{x}'), \tag{I1}$$

where $\chi_{nn}$ is the static charge susceptibility, given by the retarded Green's function as $\chi_{nn}(k) = \lim_{\omega \to 0} G^{\mathrm{R}}_{nn}(\omega, k)$, with $\omega$ a complex frequency. Due to isotropy, a generalized con-

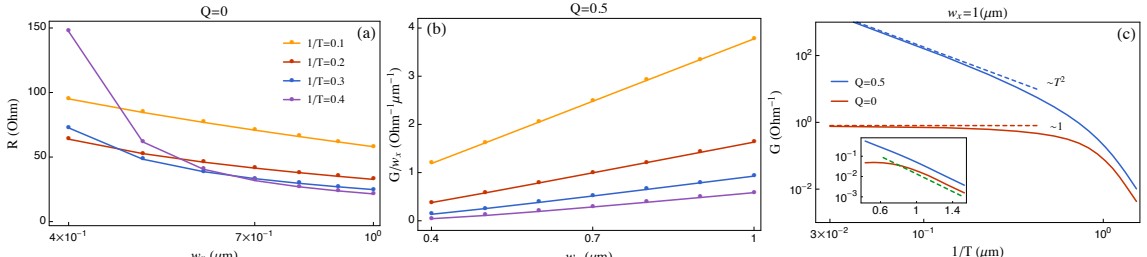

Figure 12: (a): log-linear plot of resistivity against the width $w_x$ at various $T$ at zero density. At high $T$ in the Ohmic regime, $R \sim -\log w_x$; while lowering $T$, $R$ enhances at small $w_x$ but keeps Ohmic scaling at large $w_x$ with decreased $R$. (b): conductance over width against the width at various $T$ at finite density. At high $T$ in the viscous regime, $G \sim w_x^2$; while at low $T$, $G$ approximately has the same viscous scaling but with decreased $G$. (c): log-log plot of conductance against $1/T$. At high $T$, $G \sim$ constant for zero density while $G \sim T^2$ for finite density. After decreasing $T$ below $1/T \sim w_x$, two curves coincide at the scaling $G \sim \exp(-\alpha'/T)$ ($\alpha' \approx 5$ as shown in the inset) indicating the quantum critical nature.

ductivity $\sigma_{\mu J_i}$ can be defined as

$$\sigma_{\mu J_i}(\boldsymbol{k}) = \chi_{nn}^{-1} \sigma_{nJ_i}(\boldsymbol{k}) \equiv \chi_{nn}^{-1} \lim_{\omega \to 0} \frac{G_{nJ_i}^{R}(\omega, k)}{i\omega} = \chi_{nn}^{-1} \frac{-ik_i}{k^2} \lim_{\omega \to 0} G_{nn}^{R}(\omega, k) = \frac{-ik_i}{k^2}, \quad i = x, y,$$
(I2)

where in the third step we used the current conservation Ward identity: $i\omega n = ik_i J_i$. Now, the potential difference can be determined by $V = \mu(y \gg w_{x,y}) - \mu(-y \ll -w_{x,y})$ from numerical solutions of (4).

The resulting conductance for zero density at a fixed $T$ is shown in Fig. 7(c); in the range plotted, the state is in the quantum critical regime. We find that at small width, the conductance is exponentially suppressed as the effective scattering length $1/T$ becomes larger than $w$; at larger width, the conductance grows logarithmically against the width saturating the same scaling as the Ohmic transport [31] (see Fig. 12(a)).

Schematically, we expect

$$G \sim \sigma(k = 1/w_x)$$
(I3)

(up to logarithmic corrections). In particular, this heuristic argument suggests that for ohmic transport $G \sim 1$ (close to $G \sim \log w_x$, which arises from more accurate calculation [31]), $G \sim \sqrt{n} w_x$ in a ballistic regime, $G \sim n^2 w_x^2/\eta$ in a viscous regime (in agreement with [25]), and $G \sim \exp(-\tilde{\alpha}(\mu, T)/w_x)$ in a quantum critical regime with dynamical critical exponent $z = 1$ (a new prediction of this paper).

We present the conductances of our constriction geometry across the hydrodynamic to quantum critical crossover, both at zero and finite density, in Fig. 12(a,b). Both the ohmic $G \sim \log w_x$ and viscous $G \sim w_x^2$ scaling have been reproduced by our model at high enough $T$; the exponential suppression with respect to smaller width has been confirmed in Fig. 7(c) at zero density. Upon decreasing $T$, we find that only at zero density and at large width $w_x$, the conductance is enhanced. Unlike the viscous flow, the concave current distribution for quantum critical transport does not lead to collective reductions on resistivity, but suggests a resistive "squeezed" motion in crossing the slit. Further, we estimate $G \sim \sigma_0 \sim T^0$ in the Ohmic regime, while $G \sim n^2/\eta \sim T^2$ in the viscous regime; they together enter into the quantum critical regime with $G \sim \exp(-\alpha'/T)$ as predicted by the holographic model (Fig. 12(c)).

As it is possible to measure conductances without using local imaging methods, the exponential conductance may be a simpler signature for quantum critical transport accessible in

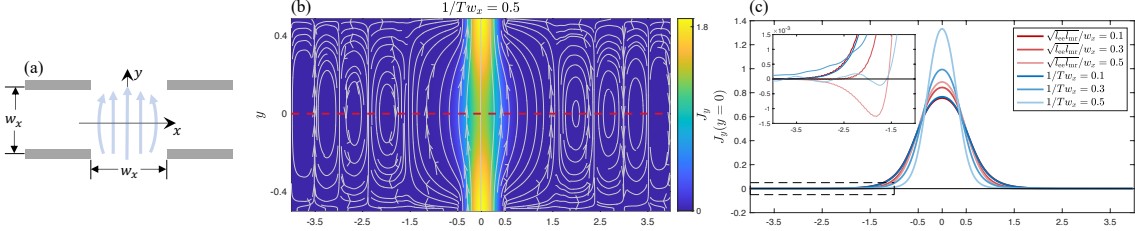

Figure 13: Vortices in a strip geometry. (a): the constriction (gray area) is located at $y = \pm w_x/2$ and $|x| \geq w_x/2$ with $w_x = 1\ \mu$m and $w_y \ll w_x$. The current enters through a slit at $y = -w_x/2$ and exists through a slit at $y = w_x/2$. (b): 2D current distribution of $J_y$ and streamlines for $1/Tw_x = 0.5$. The number of vortices would increase with increasing strip length. The tiny asymmetry is due to numerical discretization. The horizontal dashed line indicates the line at which the current distributions in (c) are calculated. (c): current distribution of $J_y$ on the line $y = 0$ for the kinetic model (red) and our holographic model at zero density (blue). The inset zooms on the backflows. By fixing $\ell_{\text{ee}} = \ell_{\text{mr}}$ but changing $\sqrt{\ell_{\text{ee}}\ell_{\text{mr}}}$, the kinetic model shows the ohmic-to-ballistic crossover. However, our holographic model indicates that the current density has a (minor) increase away from the center $x = 0$ in the intermediate regime, and begins to oscillate around zero in the quantum critical regime corresponding to the multi-vortex structure developed in (b). Note that the magnitude of the vorticity is much weaker than in the ballistic regime.

experiment.

## J  Vorticity in a strip geometry

In this section we study the quantum transport in a different strip geometry [53, 56, 57] (Fig. 13(a)). Note that our numerical codes allow us to simply change the region I in numerics and use the same $\sigma(k)$ which we used for the constriction geometry.

The current distribution of a quantum critical flow is shown in Fig. 13(b). We find that vortices are developed due to the nonlocal $k$-dependence of the conductivity; however, the quantum critical flow shows distinctive behavior when compared to either ohmic, ballistic or viscous flow [53]. In Fig. 13(c), we compare our holographic model at zero density to the kinetic model discussed in Appendix G. We can estimate that $\sqrt{\ell_{\text{ee}}\ell_{\text{mr}}}$ is the underlying length scale for transport, as is the Planckian length $1/T$ in the quantum critical model.

Note that both models exhibit an ohmic transport regime on long length scales, and therefore when $w_x$ is large, the current distributions look very similar. Upon decreasing $w_x$ relative to either $\sqrt{\ell_{\text{ee}}\ell_{\text{mr}}}$ or $1/T$, the two models will enter into different regimes of transport, and correspondingly the current distributions appear rather different. The kinetic model displays the ohmic-to-ballistic crossover whenever $\ell_{\text{ee}} = \ell_{\text{mr}}$ (which we have assumed), and will develop one vortex in the ballistic regime. On the other hand, our holographic model has an unusual (minor) increment of the current away from the center in the intermediate regime, then the current starts to oscillate around the zero point in the quantum critical regime. Such oscillation, with negative currents indicating backflows against applied field, induces muti-vortex structure in Fig. 13(b). Yet, the vorticity strength in the quantum critical regime is relatively weaker than that of a ballistic flow, which could be an experimental signature of the difference between the ohmic-to-ballistic crossover versus an ohmic-to-quantum critical crossover.

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
