# Peer review of "Fingerprints of quantum criticality in locally resolved transport"

_SciPost Physics, doi:SciPost Phys. 13, 004 (2022)_

## Round 3 · Referee Report · Anonymous · 2022-4-7

Report

The manuscript argued that novel experimental technique which enables measurement of local current density can detect signatures of quantum critical transport. As a demonstration, the authors calculated the current profile in a constriction geometry under various transport mechanisms: ohmic, ballistic, viscous and quantum critical. The new contribution of the manuscript is to demonstrate that within the quantum critical regime, $\sigma(k)\sim\exp(-\alpha k/T)$ and the current profile inside the constriction is sinusoidal. The authors also compared their holographic theory of charge-neutral transport with experiments in graphene and found reasonable agreement. I think the content of the manuscript is interesting and recommend it for publication after the authors addresses the comments below.

1. In Eqs.(2) and (3), the authors separated the electric field into a homogeneous external field and a response field $\tilde{E}_i$ due to boundary, and they proposed an ansatz such that $\tilde{E}_i=0$ outside of the constriction wall. This ansatz is mathematically sufficient for calculating the current profile but physically misleading because it have omitted the field due to bulk charge distribution $-\partial_i\mu$. Because the conductivity tensor is transverse, the bulk charge field is a zero mode of Eq.(1) and therefore not effective in current profile computation, but it is crucial for computation of total conductance. The authors should clarify their derivation of Eqs. (2) and (3).

2. In the comparison to graphene experiment, it is unclear to me how the planckian length scale in Eq.(10) is related to the paramters in the theory. For example, how is $\alpha$ in Eq.(8) related to Eq.(10)? Please clarify.

The following two questions are for my personal curiosity:

3. The author's argument for sinusoidal current profile is based on heuristics and numerical data. The current profiles for the ohmic and the viscous regime are obtained by projecting Eq.(1) onto the constriction line and solving the resulting 1D integral equation for the current density. Given the analytic simplicity of the quantum critical conductivity in Eq.(8), I am wondering whether similar procedures can be done in this case and obtain an analytic solution?

4. In studies of ohmic-ballistic-viscous cross over, it was found that the resistance of the system can be approximated as $R=R_{ohmic}+1/(G_{ballistic}+G_{viscous})$ where $G$ denotes the conductances. Do you expect similar epxressions to hold in quantum critical transport?

---

## Round 3 · Referee Report · Anonymous · 2022-4-20

Strengths

1 - Extends the gamut of experimental observables sensitive to quantum critical (QC) behaviour by considering the local (generalized) conductivities of 2d QC systems.

2 - Setup with minimal assumptions; Allows for great theoretical as well as experimental control through only a couple of parameters.

3 - The predicted features of the local conductivity in the QC regime are heuristically a consequence of conformal invariance alone. Hence they can be expected to be universal.

4- Results match with existing experiments in graphene after fixing a single input parameter.

Weaknesses

1 - Parameter regime necessary to measure QC transport in this setup unlikely to be reached in experiment (in graphene) due to the presence of charge puddles and the difficulty of manufacturing sub-micron constrictions.

2 - Analysis restricted to 2d quantum critical systems.

3 - List of transport regimes considered is not exhaustive. "Superdiffusive" conductivity found in [Phys. Rev. B 102 245434 (2020)] not considered.

Report

The main message of the present paper is that locally resolved transport holds information that is distinctly quantum critical (QC). In particular the authors used the local conductivity and the corresponding current flowing in a 2d channel with a constriction to make their case.

I find the results as well as the general message presented in the paper well substantiated and of great importance for future transport experiments in the QC regime. I believe, however, there are a few points that need to be addressed before I can recommend the paper for publication in this journal:

1) In recent years, the superdiffusive (SD) transport regime has been identified in 2d Dirac materials at charge neutrality [Phys. Rev. Lett. 123, 195302 (2019)]. It has been further shown that within the SD regime, the conductivity predicted by hydrodynamics becomes wavenumber dependent [Phys. Rev. B 102, 245434 (2020)].

I think the authors should comment on whether the SD regime is a relevant transport regime to consider in local imaging experiments. If yes, how does the current flow in that regime compare to the flow in the QC one?

2) In order to reach the QC regime in a graphene channel with a constriction of order $1 {\mathrm{\mu m}}$ , the temperature of the graphene sample must be $T = {\cal O}(1 {\rm K})$ . At such low temperatures, however, the effects of charge puddles becomes important [ J. Phys.: Condens. Matter 30 053001 (2018)].

It would be interesting to see a discussion/comment on whether the charge puddles affect the analysis found in the paper, especially around the charge neutrality point. In particular, is there a lower temperature limit set by charge puddles and if yes, does it forbid us from exploring the QC regime?

3) One way to side-step the charge puddle issue would be to use 3d QC metals. Because of this, I think it would be useful to comment on how the results in the paper depend on the dimensionality of the system, if at all.

Some further, minor points I would like to bring to the attention of the authors are the following:

i) Some of the cross-references between objects in the paper (tables, equations, sections) are wrong or not link to the proper place.

ii) It seems the authors decided to present their main point in a short letter form and relegate everything else in appendices. I believe the paper would be clearer if some of the appendices were pulled up to the main text e.g. appendix A.

Requested changes

1 - Explain why the superdiffusive regime of Phys. Rev. Lett. 123 (2019) is/isn't relevant for the purposes of the paper.

2 - Explain how charge puddles may affect the conclusions of the paper and the ability to measure charge transport in QC graphene.

3 - Discuss how the results are expected to depend on the dimensionality of the material.

3 - Fix cross-referencing issues (e.g. table 1 is referred to as table 6 near the end of the main text).

4 - (Optional) Incorporate parts of the appendices into the main text to increase the paper's clarity.

---

## Round 4 · Referee Report · Anonymous · 2022-5-13

Report

The authors have addressed my questions and I recommend for publication.

---

## Round 4 · Referee Report · Anonymous · 2022-5-19

Report

I believe the authors have given sufficient responses to my remarks. As such, I can now suggest their manuscript for publication in this journal.

---

## Round 4 · Author Response

Errors in user-supplied markup (flagged; corrections coming soon)

Apologies that we had prepared a nice LaTeXed PDF with our reply but I cannot figure out how to upload it... so I have cut and paste and some math formulas are a little annoying to read.

Response to Referee 1:

"Explain why the superdiffusive regime of Phys. Rev. Lett. 123 (2019) (also Phys. Rev. B 102, 245434 (2020)) is/isn't relevant for the purposes of the paper."

Note that these papers are Ref.[34,35] in our paper. The main purpose of our paper is to show that the sinusoidal current profile is a fingerprint for quantum critical flow -- at least near certain quantum critical points, including those coming from holographic models -- and is distinguishable from hydrodynamic flows.

In the interesting papers mentioned by the referee, evaluating the explicit formulas for $\sigma(k,\omega)$ given there lead to, in the $\omega \rightarrow 0$ limit, the form $\sigma\sim 1/(1+v^2\tau_{c,1}\tau_{c,2}q^2)$ when $q<(v\tau_{c,2})^{-1}$, so they can interpret between $\sigma\sim 1$ ($q\ll (v\tau_{c,1})^{-1}$) and $\sigma\sim 1/q^2$ ($q\gg (v\tau_{c,1})^{-1}$). This will look quite a lot like viscous flow, and we suppose this comes from the emergence of approximate momentum conservation of electron and hole fluids separately. We have added a few sentences to Section 6 of the main text commenting on this interesting observation. But we also note there that this behavior is quite different from the quantum critical flow, $\mathrm{Re}~\sigma\sim \exp(-\gamma k)$.

One can certainly do an analysis of experimental data using this kinetic theory instead. One merit of our approach is that we only required a \emph{one-parameter fitting} between our holographic model and the experimental data, which already led a good fit. Moreover, our estimated fit parameter agrees quantitatively with another independent experiment. Moreover, our approach is capable of interpreting between the hydrodynamic (Ohmic, in the case of charge-neutral systems) and quantum critical regimes.
If one wants to analyze the quantum critical regime using approaches based on kinetic theories, the breakdown of semiclassical dynamics on length scales smaller than $\hbar v_{\mathrm{F}}/k_{\mathrm{B}} T$ must be accounted for quite carefully. This could be an interesting future project.

"In order to reach the QC regime in a graphene channel with a constriction of order $1\;\mu$m, the temperature of the graphene sample must be $T=\mathcal{O}(1K)$. At such low temperatures, however, the effects of charge puddles becomes important. Explain how charge puddles may affect the conclusions of the paper and the ability to measure charge transport in QC graphene."

First, we want to point out that in order to reach the quantum critical regime with a constriction of width $1\;\mu$m, the temperature could be $T \lesssim 40$ K, instead of $T \lesssim 1$ K. We found this by simply plugging in numbers into our Eq. (11) to find the Planckian length scale.

In experiments on graphene samples using hBN substrates (the default method used today) one typically finds that the amplitude of charge puddles is around 30 K, and their size is around 100 nm. Since both the size of charge puddles is small relative to the device sizes we are advocating using, and it is possible to enter the quantum critical regime at temperatures above 30 K, we believe it is definitely possible to image these phenomena in experiment if our theory is accurate. We emphasize that in Fig. 4 in the main text, we evaluated (albeit in a model neglecting charge puddles) and saw that for 600 nm constrictions and $T\sim 128\;$ K, it should be possible to see quite strong deviations from Ohmic or viscous flow patterns! And given the numbers highlighted above, this seems entirely reasonable to try.

We have added a few comments in the main text discussing these points and explaining the relevant length/energy scales of charge puddles. We thank the referee for the suggestion.

"Discuss how the results are expected to depend on the dimensionality of the material."

From the perspective of holographic duality, we expect the exponential decaying conductivity $\mathrm{Re}~\sigma\sim \exp(-\alpha k)$ to be held in any dimension since the spectral weight will always decay exponentially toward the horizon. A field theory analysis similar to Ref.[29] would also suggest that the quantum critical sinusoidal profile is insensitive to spatial dimensions. Of course, the imaging procedure would be quite difficult in a three dimensional model, so we expect our ideas will be most interesting in two-dimensional systems.

=======

Response to Referee 2:

"This ansatz is mathematically sufficient for calculating the current profile but physically misleading because it have omitted the field due to bulk charge distribution $-\partial_i\mu$. Because the conductivity tensor is transverse, the bulk charge field is a zero mode of Eq.(1) and therefore not effective in current profile computation, but it is crucial for computation of total conductance. The authors should clarify their derivation of Eqs. (2) and (3)."

Our Eqs.(1), (2) and (3) aim to compute spatially resolved current profiles through $k$-dependent conductivities. We agree that these formulae do not capture the bulk charge distribution, therefore, we explained in Appendix I that a generalization of Eq.(1) to spatially resolved charge density $n(x)$ is required to compute the conductance $G$. A detailed discussion of this issue is provided in Appendix I. Nevertheless, to help direct the reader's attention to the ease of this generalization, we have moved one equation (now Eq.(4)) and some discussion from Appendix I into the main text in Section 2.

"In the comparison to graphene experiment, it is unclear to me how the planckian length scale in Eq.(11) is related to the paramters in the theory. For example, how is $\alpha$ in Eq.(9) related to Eq.(11)? Please clarify."

At a fixed temperature $T$, our holographic model has no fit parameters. $\alpha$ is an $\mathcal{O}(1)$ number whose exact value needs to be determined numerically as discussed in Appendices E and F. To compare to experiment, we need to fix the number $C$. In the holographic model using natural units, we have $C=1$. We run the holographic simulation for a fixed width device at various $T$ (as its output depends only on the product $Tw$, in dimensionless units), generate the current flow patterns, and find which one is the best fit to the experimental data. That gives us $T_{\mathrm{fit}}$ (which assumes $C=1$). However, we find $T_{\mathrm{fit}}$ is not equal to the experimental temperature $T_{\mathrm{exp}}$ but they satisfy $1/T_{\mathrm{exp}}w_x\propto 1/T_{\mathrm{fit}}w_x$. Hence, according to Eq.(11), we can determine $C$ through $1/T_{\mathrm{exp}}w_x=C\times 1/T_{\mathrm{fit}}w_x$.

We have added a sentence explaining this to the manuscript in the experimental section and hope this clarifies the issue.

"The author's argument for sinusoidal current profile is based on heuristics and numerical data. The current profiles for the ohmic and the viscous regime are obtained by projecting Eq.(1) onto the constriction line and solving the resulting 1D integral equation for the current density. Given the analytic simplicity of the quantum critical conductivity in Eq.(9), I am wondering whether similar procedures can be done in this case and obtain an analytic solution?"

Unfortunately, we were unable to find analytic expressions for the quantum critical current profile when we tried to solve these integral equations. However, we believe that the sinusoidal profile captures the main feature of the quantum critical flow, at least around the center of the constriction. The obstacle to get a closed form analytical expression is in part fixing boundary conditions, but since the experimental data has limited accuracy near the boundary due to finite resolution, the universal feature near the center would likely be more important. But it could be interesting if future authors are capable of solving this problem!

"In studies of ohmic-ballistic-viscous cross over, it was found that the resistance of the system can be approximated as $R=R_{ohmic}+1/(G_{ballistic}+G_{viscous})$, where $G$ denotes the conductances. Do you expect similar epxressions to hold in quantum critical transport?"

Our holographic model is capable of showing the breakdown of Matthiessen's rule in the hydrodynamic regime (see Eqn.(10) and Eq.(F6)). But we doubt that, for example, one could just replace $G_{ballistic}$ or $G_{viscous}$ with $G_{qc}$ in a quantum critical regime; the precise mathematical formula the referee quotes is probably rather bit special to the models studied in earlier papers.

---

## Round 4 · List of Changes

The changes we made to the manuscript are explained in the responses above.

---

## Editorial Decision

published